# Co-infection with Chikungunya virus alters trafficking of pathogenic CD8[+] T cells into the brain and prevents *Plasmodium*-induced neuropathology

Teck-Hui Teo[1], Shanshan W Howland[1], Carla Claser[1], Sin Yee Gun[1], Chek Meng Poh[1,‡], Wendy WL Lee[1], Fok-Moon Lum[1] (iD), Lisa FP Ng[1,2,*,†] (iD) & Laurent Rénia[1,**,†] (iD)

## Abstract

Arboviral diseases have risen significantly over the last 40 years, increasing the risk of co-infection with other endemic disease such as malaria. However, nothing is known about the impact arboviruses have on the host response toward heterologous pathogens during co-infection. Here, we investigate the effects of Chikungunya virus (CHIKV) co-infection on the susceptibility and severity of malaria infection. Using the *Plasmodium berghei* ANKA (PbA) experimental cerebral malaria (ECM) model, we show that concurrent co-infection induced the most prominent changes in ECM manifestation. Concurrent co-infection protected mice from ECM mortality without affecting parasite development in the blood. This protection was mediated by the alteration of parasite-specific CD8[+] T-cell trafficking through an IFNγ-mediated mechanism. Co-infection with CHIKV induced higher splenic IFNγ levels that lead to high local levels of CXCL9 and CXCL10. This induced retention of CXCR3-expressing pathogenic CD8[+] T cells in the spleen and prevented their migration to the brain. This then averts all downstream pathogenic events such as parasite sequestration in the brain and disruption of blood–brain barrier that prevents ECM-induced mortality in co-infected mice.

**Keywords** CD8[+] T-cell trafficking; Chikungunya; co-infection; malaria
**Subject Categories** Immunology; Microbiology, Virology & Host Pathogen Interaction

## Introduction

With the advent of new and efficient methods to identify and diagnose known and emerging pathogens, it is evident that co-infections are a common phenomenon. Particularly, co-infection with more than one pathogen such as HIV, *Mycobacterium tuberculosis*, hepatitis viruses, helminths, and *Plasmodium* has been estimated to affect about one-third of the human population in developing countries (Stelekati & Wherry, 2012). Existing epidemiological data suggest a greater incidence of negative effects on pathogen-specific host immune responses during co-infection (Stelekati *et al*, 2014). However, the underlying mechanisms remain poorly understood (Stelekati & Wherry, 2012).

Among the prevalent infectious diseases in the world, mosquito-borne parasites and viruses are frequently co-endemic in intertropical regions. Malaria transmission still persists in 95 countries, accounting for over 214 million new cases worldwide in 2015 alone (World Health Organization (WHO), 2015). Arthropod-borne viruses (arboviruses) such as Chikungunya virus (CHIKV), Zika virus (ZIKV), and dengue virus are also endemic and co-endemic with malaria in many tropical countries such as Africa, Latin America, and Asia (Power, 2015; WHO, 2015). This occurrence increases the risks of co-infection in these populations. Classical clinical symptoms of arboviruses include febrile illness, rashes, and joint pain. Thus, generic febrile symptoms in arbovirus-infected patients make clinical identification of their co-infection with malaria parasites difficult.

Among the arboviruses, co-infection of *Plasmodium* with CHIKV has been reported in patients (Hertz *et al*, 2012; Baba *et al*, 2013; Chipwaza *et al*, 2014; Ayorinde *et al*, 2016; Waggoner *et al*, 2017). In one study, patients were reported to concomitantly harbor *P. falciparum* parasites (diagnosed by microscopy) and CHIKV (diagnosed by PCR; Hertz *et al*, 2012). In two other studies, patients acutely infected with malaria were positive for CHIKV-specific IgM (Ayorinde *et al*, 2016) or possessed neutralizing CHIKV-specific antibodies in their sera (Baba *et al*, 2013). While these studies demonstrate that infection by *Plasmodium* parasites and CHIKV could occur together, the impact of co-infection on host susceptibility and the respective infection-induced pathologies remain

1  Singapore Immunology Network, Agency for Science, Technology and Research (A*STAR), Singapore, Singapore
2  Institute of Infection and Global Health, University of Liverpool, Liverpool, UK
  *Corresponding author. Tel: +65 64070028; Fax: +65 64642057; E-mail: lisa_ng@immunol.a-star.edu.sg
  **Corresponding author. Tel: +65 64070005; Fax: +65 64642056; E-mail: renia_laurent@immunol.a-star.edu.sg
  †These authors contributed equally to this work as senior authors
  ‡Present address: Center of Influenza Research and School of Public Health, The University of Hong Kong, Hong Kong, China

unknown. Both diseases induce strong but different dynamics in innate and adaptive immune responses (Boubou *et al*, 1999; Renia *et al*, 2006; Chirathaworn *et al*, 2010; Gardner *et al*, 2010; Hoarau *et al*, 2010; Chaaitanya *et al*, 2011; Claser *et al*, 2011; Kelvin *et al*, 2011; Wauquier *et al*, 2011; Hansen, 2012; Rénia *et al*, 2012; Rovira-Vallbona *et al*, 2012; Boef *et al*, 2013; Teo *et al*, 2013). Thus, co-infection may modify both kinetics and characteristics of pathogen-specific immunity that could alter disease outcome, that is, protection versus immunopathology.

In this study, we explore the effects of CHIKV co-infection on the susceptibility and severity of malaria infection using a mouse model of experimental cerebral malaria (ECM) induced by *P. berghei* ANKA (PbA).

## Results

### Concurrent co-infection with CHIKV infection protects mice from ECM

Different scenarios of co-infection between CHIKV and PbA were investigated (Fig 1). In the well-established PbA-ECM model, PbA infection typically results in 70–80% ECM-induced death in mice between 6 and 12 days post-infection (dpi; Engwerda *et al*, 2005). Interestingly, concurrent co-infection with CHIKV protected mice from death despite similar parasitemia levels during the ECM window period of 6–12 dpi (Fig 1A). Instead, mice died of hyper-parasitemia much later at 28 dpi (Fig 1A). Contrastingly, there was no change in the occurrence of ECM in mice infected with PbA either 4 days before (Fig 1B), 4 days after (Fig 1C), or 15 days after CHIKV infection (Fig 1D). There was also no change in parasitemia levels in all sequential infection scenarios (Fig 1B and D), except for higher parasitemia just prior to and during the ECM window when CHIKV was inoculated 4 days before PbA infection (Fig 1C).

### Co-infection reduces parasite sequestration in the brain and preserves the blood–brain barrier (BBB)

One vital characteristic of ECM is sequestration of parasites in the brain during neurological manifestation (Amante *et al*, 2010; Claser *et al*, 2011; Howland *et al*, 2013). The PbA clone used in this study has been genetically modified to express luciferase (Amante *et al*, 2007) that allows for assessment of parasite sequestration kinetics

in the deep tissues using bioluminescent imaging. Compared to PbA infection alone, mice concurrently infected with both pathogens had significantly reduced bioluminescence signals in the whole body and in the head at 5 dpi (Fig 2A and B). This time point also coincided with the time when singly PbA-infected mice would normally present mild neurological manifestations. At later time points, differences in parasite sequestration were not observed (Fig 2A and B). To specifically quantify the parasites that sequestered accurately in the brain, mice were perfused at 6 dpi and the *ex vivo* bioluminescence signals were recorded from the brains. Expectedly, concurrent co-infection reduced the parasite load in the isolated brains at 6 dpi (Fig 2C).

Experimental cerebral malaria mortality is due to a breach of the blood–brain barrier (BBB) (Thumwood *et al*, 1988; van der Heyde *et al*, 2001). Blood–brain barrier permeability can be quantified by intravenous injection of Evans blue dye, as this dye is normally excluded from the brain parenchyma unless the BBB is ruptured (Clasen *et al*, 1970). Evans blue permeability assay was performed when PbA-infected mice began to display neurological symptoms at 6 dpi. Dye levels were significantly higher in all brains compared to the brains of co-infected mice. As shown by the reduction of dye recovered from the perfused brain, concurrent co-infection preserved BBB integrity (Fig 2D). Taken together, protection from ECM lethality by concurrent CHIKV co-infection is associated with a reduction in parasite sequestration in the brain and preservation of BBB integrity.

### Co-infection prevents brain endothelium cross-presentation of parasite-derived antigens

When ECM-susceptible mice develop neurological signs, leukocytes and in particular parasite-specific CD8[+] T cells accumulate in the brain (Belnoue *et al*, 2002; Howland *et al*, 2013). These CD8[+] T cells have been shown to recognize parasite antigens cross-presented by activated brain endothelium (Belnoue *et al*, 2002; Howland *et al*, 2013, 2015b) and mediate the disruption of the BBB during PbA infection (Howland *et al*, 2015a). Since cross-presentation of parasite antigens is dependent on the parasite load in the brain (Howland *et al*, 2013), and a reduced parasite load was observed upon co-infection (Fig 2C), we assessed whether this influenced the degree of parasite antigens cross-presentation in the brain endothelium of co-infected mice. Using a reporter cell line to measure the level of *in vivo* cross-presentation of an immunodominant "Pb1" parasite epitope by brain endothelial cells (Howland

---

**Figure 1.  Concurrent co-infection with CHIKV and PbA protects mice against ECM.**

A  Parasitemia and mortality curve of PbA (*n* = 7) and PbA + CHIKV (*n* = 7) groups. PbA and CHIKV infection were given concurrently. Parasitemia data were analyzed by Mann–Whitney two-tailed analysis (4 dpi; ***$P$ = 0.0006). Mortality curve was analyzed by log-rank (Mantel–Cox) test (**$P$ = 0.0081).

B  Parasitemia and mortality curve of PbA (*n* = 6) and PbA (4 dpi) + CHIKV (*n* = 7) groups. CHIKV infection was given on 4 days post-PbA infection. Mortality curve was analyzed by log-rank (Mantel–Cox) test (ns: $P$ = 0.7425).

C  Parasitemia and mortality curve of PbA (*n* = 7) and PbA + CHIKV (4 dpi) (*n* = 5) groups. PbA infection was given on 4 days post-CHIKV infection. Parasitemia data were analyzed by Mann–Whitney two-tailed analysis (3 dpi; **$P$ = 0.0025, 4 dpi; **$P$ = 0.0025, 5 dpi; *$P$ = 0.0101, 6 dpi; *$P$ = 0.0177). Mortality curve was analyzed by log-rank (Mantel–Cox) test (ns: $P$ = 0.1180).

D  Parasitemia and mortality curve of PbA (*n* = 5) and CHIKV recovered + PbA (*n* = 4) groups. PbA infection was given on day 15, after mice recovered from CHIKV-induced joint-swelling on day 14. Mortality curve was analyzed by log-rank (Mantel–Cox) test (ns: $P$ = 0.2923).

Data information: Shaded region within the graphs represents the ECM time window. Schematics of infection schedule are shown above each figure panel. Data shown are representative of two independent experiments. Parasitemia data are represented as mean ± SD.

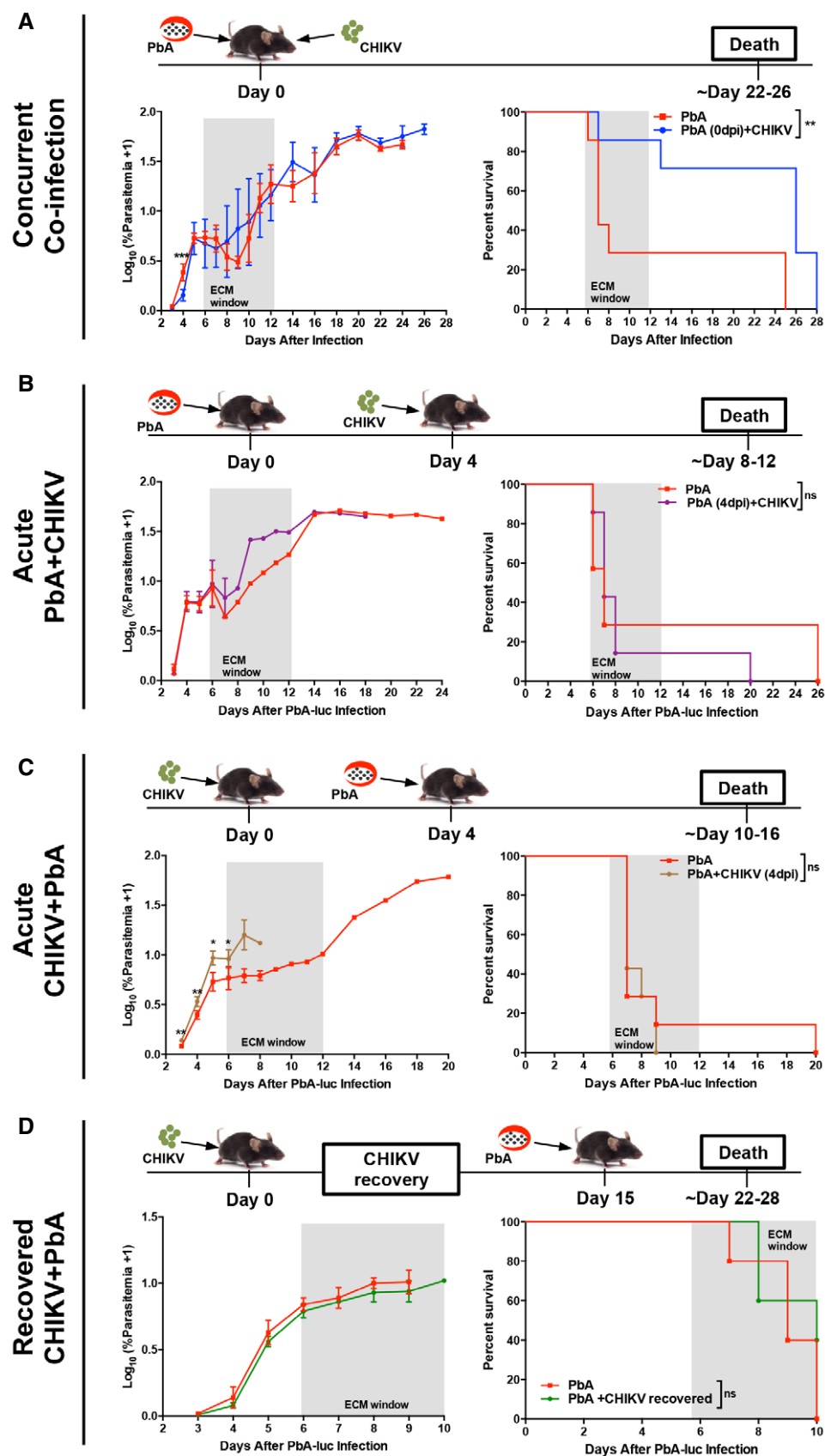

Figure 1.

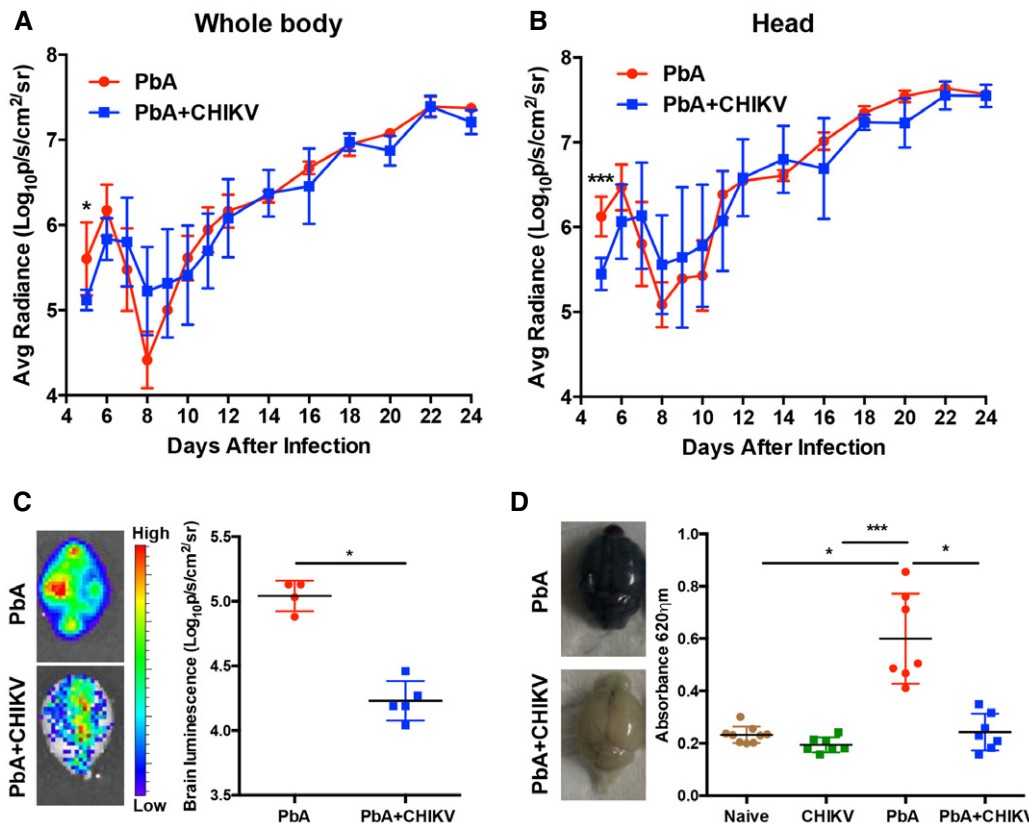

**Figure 2. Concurrent co-infection prevents sequestration of parasites and BBB permeability in the brain.**

A, B Parasite load in the whole body and head of PbA ($n = 7$) and PbA + CHIKV ($n = 7$) groups. Data shown were representative of two independent experiments. Mann–Whitney two-tailed analysis (whole body—5 dpi; *$P = 0.0379$, head—5 dpi; ***$P = 0.0006$).

C *Ex vivo* parasite load in the brain of PbA ($n = 4$) and PbA + CHIKV ($n = 5$) groups on 6 dpi. Representative pseudoimages from each group are shown. Mann–Whitney two-tailed analysis (*$P = 0.159$).

D Evans blue permeability assay for BBB in naïve ($n = 9$), CHIKV only ($n = 7$), PbA ($n = 7$), and PbA + CHIKV ($n = 7$) on 6 dpi. Representative images of the brain after Evans blue injection from infected groups are shown. Data shown were pooled from two independent experiments. Kruskal–Wallis test with Dunn's multiple comparison test (naïve versus PbA; *mean rank diff = −13.06, CHIKV versus PbA; ***mean rank diff = −19.36, PbA versus PbA + CHIKV; *mean rank diff = −13.14).

Data information: All data are expressed as mean ± SD.

*et al*, 2013), we showed that co-infection greatly reduced cross-presentation (Fig 3A).

## Co-infection prevents sequestration of pathogenic CD8[+] T cells in the brain

A lower level of recruitment of CD45[+] leukocytes into the brain was observed during co-infection (Fig 3B). Among these CD45[+] cells, both total and activated (LFA-1[+]) CD4[+] T cells and CD8[+] T cells were significantly less in the co-infected mice at 6 dpi (Fig 3C and D). Particularly, co-infection was demonstrated to reduce the accumulation of CD8[+] T cells specific for a highly immunogenic parasite CD8[+] T-cell epitope, Pb1 (Howland *et al*, 2013), in the brain (Fig 3E).

## Co-infection does not impair priming and expansion of functional CD8[+] T cells in the spleen

Reduced accumulation of CD8[+] T cells in the brain could result from impaired priming and expansion in the spleen, or altered

T-cell trafficking to the brain. To address the first scenario, spleens were isolated at 6 dpi and profiled for the level of induced T cells. Surprisingly, it was observed that concurrent co-infection with virus enlarged the spleen even more than single infection with PbA (Fig 4A). This phenomenon was associated with higher numbers of total splenocytes in the co-infected mice (Fig 4A). Phenotyping by flow cytometry revealed that more activated total (LFA-1[+]) CD4[+] T cells and CD8[+] T cells were present in the enlarged spleens of co-infected mice (Fig 4B and C). Similarly, the levels of Pb1-specific CD8[+] T cells in the spleen were also higher in the co-infected mice than in single PbA-infected mice (Fig 4D). To confirm that the CD8[+] T cells induced in the spleen of co-infected mice are functional, an *in vivo* cytolysis assay was performed. In both single PbA-infected and co-infected mice, > 95% of transferred Pb1-pulsed naïve splenocytes were eliminated (Fig 4E), demonstrating that CD8[+] T cells induced in the spleens of co-infected mice are cytolytic. These results suggest that co-infection does not impair the host's ability to generate functional T cells in the spleen.

**Figure 3.  Concurrent co-infection prevents T-cell sequestration and microvessel cross-presentation in the brain.**

A       Brain microvessel cross-presentation for Pb1 epitopes in naïve (*n* = 5), PbA (*n* = 5) and PbA + CHIKV (*n* = 5) groups on 6 dpi. One-way ANOVA with Tukey's post-test (naïve versus PbA; **mean diff* = −922.8, PbA versus PbA + CHIKV; **mean diff* = 888.2).

B–D    CD45[+], total and LFA-1[+]CD4[+] T cells, and total and LFA-1[+]CD8[+] T cells in the brain of naïve (ochre, *n* = 8), CHIKV (green, *n* = 7), PbA (red, *n* = 6), and PbA + CHIKV (blue, *n* = 5) groups on 6 dpi. Data shown were representative of two independent experiments. All data were analyzed by one-way ANOVA with Tukey's post-test. For CD45[+] cells (B): naïve versus PbA; **mean diff* = −142,082, CHIKV versus PbA; **mean diff* = −145,113, PbA versus PbA + CHIKV; **mean diff* = 120725. For total CD4[+] T cells (C, left): naïve versus PbA; ***mean diff* = −12,000, CHIKV versus PbA; **mean diff* = −11,360. For LFA-1[+]CD4[+] T cells (C, right): naïve versus PbA; ***mean diff* = −7,828, CHIKV versus PbA; **mean diff* = −7,243, PbA versus PbA + CHIKV; **mean diff* = 4,521. For total CD8[+] T cells (D, left): naïve versus PbA; ***mean diff* = −65,880, CHIKV versus PbA; ***mean diff* = −64,730, PbA versus PbA + CHIKV; **mean diff* = 46,430. For LFA-1[+]CD8[+] T cells (D, right): naïve versus PbA; ***mean diff* = −57,400, CHIKV versus PbA; ***mean diff* = −56,280, PbA versus PbA + CHIKV; **mean diff* = 39,320.

E       Parasite epitope (Pb1)-specific CD8[+] T cells in the brain of PbA (*n* = 6) and PbA + CHIKV (*n* = 5) groups on 6 dpi. Data shown were representative of two independent experiments. Mann–Whitney two-tailed analysis (**P* = 0.0303).

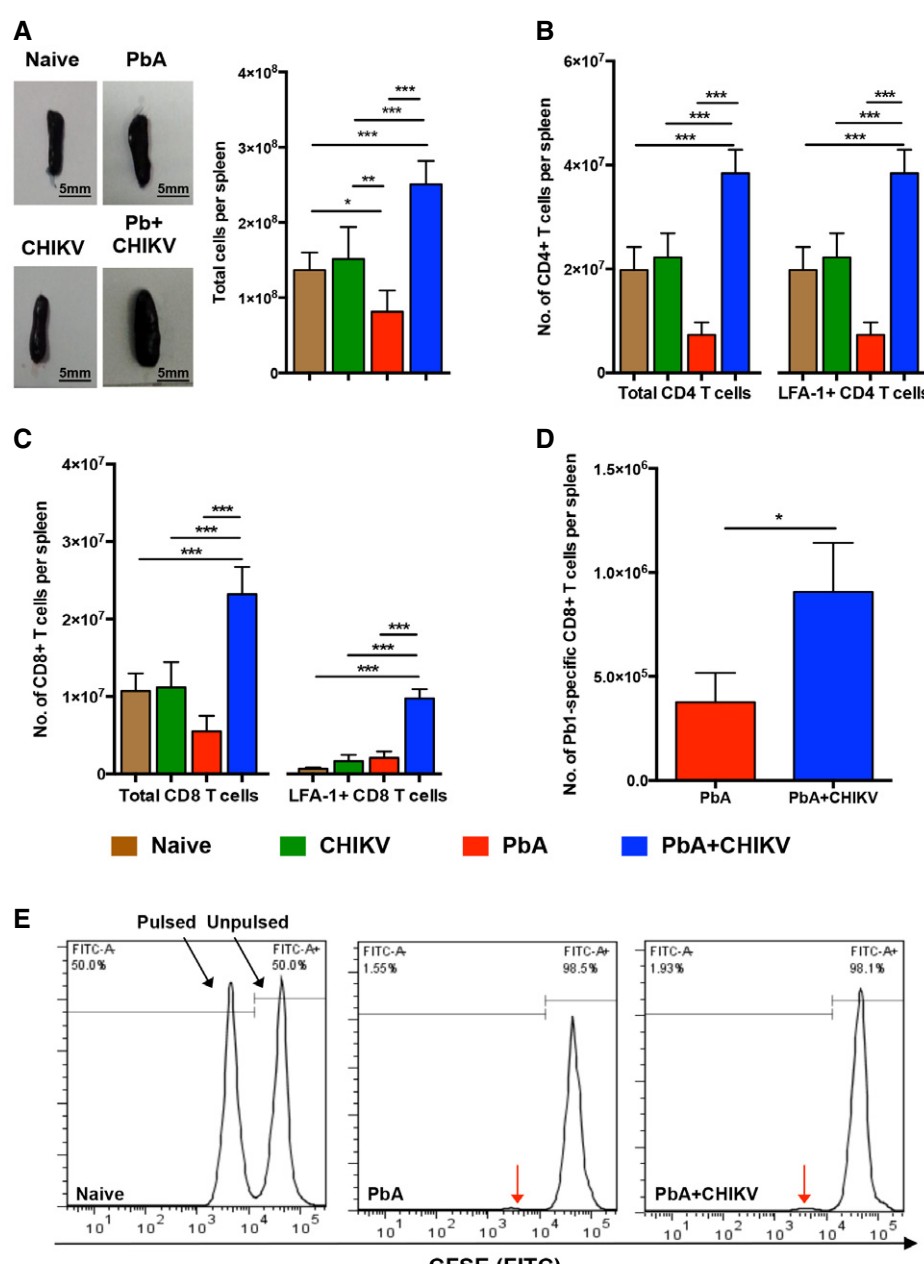

**Figure 4. Normal priming and expansion of functional T cells in the spleen during concurrent co-infection.**

A–C   Total splenocytes, total and LFA-1$^+$CD4$^+$ T cells, and total and LFA-1$^+$CD8$^+$ T cells in the spleen of naïve ($n = 8$), CHIKV ($n = 7$), PbA ($n = 6$), and PbA + CHIKV ($n = 6$) groups on 6 dpi. (A) Representative images of spleens isolated on 6 dpi showing spleen enlargement during concurrent co-infection are shown. Total splenocytes were determined by live cell count with Trypan blue staining. Data shown are representative of two independent experiments. All data were analyzed by one-way ANOVA with Tukey's post-test. For total splenocytes (A): naïve versus PbA; *mean diff = 5.5 × 10$^7$, naïve versus PbA + CHIKV; ***mean diff = −1.142 × 10$^8$, CHIKV versus PbA; **mean diff = 6.993 × 10$^7$, CHIKV versus PbA + CHIKV; ***mean diff = −9.924 × 10$^7$, PbA versus PbA + CHIKV; ***mean diff = −1.692 × 10$^8$. For total CD4$^+$ T cells (B, left): naïve versus PbA + CHIKV; ***mean diff = −1.866 × 10$^7$, CHIKV versus PbA + CHIKV; ***mean diff = −1.621 × 10$^7$, PbA versus PbA + CHIKV; ***mean diff = −3.11 × 10$^7$. For LFA-1$^+$CD4$^+$ T cells (B, right): naïve versus PbA + CHIKV; ***mean diff = −1.056 × 10$^7$, CHIKV versus PbA + CHIKV; ***mean diff = −8.321 × 10$^6$, PbA versus PbA + CHIKV; ***mean diff = −1.119 × 10$^7$. For total CD8$^+$ T cells (C, left): naïve versus PbA + CHIKV; ***mean diff = −1.250 × 10$^7$, CHIKV versus PbA + CHIKV; ***mean diff = −1.203 × 10$^7$, PbA versus PbA + CHIKV; ***mean diff = −1.770 × 10$^7$. For LFA-1$^+$CD8$^+$ T cells (C, right): naïve versus PbA + CHIKV; ***mean diff = −9.094 × 10$^6$, CHIKV versus PbA + CHIKV; ***mean diff = −8.109 × 10$^6$, PbA versus PbA + CHIKV; ***mean diff = −7.687 × 10$^6$.

D   Parasite epitope (Pb1)-specific CD8$^+$ T cells in the spleen of PbA ($n = 4$) and PbA + CHIKV ($n = 5$) groups on 6 dpi. Data shown are representative of two independent experiments. Mann–Whitney two-tailed analysis (*$P = 0.0159$).

E   Representative plot of in vivo cytotoxic assay of naive ($n = 2$), PbA ($n = 4$) and PbA + CHIKV ($n = 7$). Equal numbers of CFSE$^{hi}$ unpulsed naive splenocytes and CFSE$^{lo}$ SQLLNAKYL (Pb1) pulsed splenocytes were transferred into the mice on 6 dpi. All mice from PbA and CHIKV + PbA groups showed > 95% of specific lysis on Pb1 pulsed donor cells. Red arrows indicate pulsed splenocytes that were lysed.

Data information: All bar graphs are expressed as mean + SD.

   

## Concurrent co-infection reduces the migration capacity of CD8[+] T cell to the brain and suppresses surface expression of CXCR3 and CD43

The higher number of functional CD8[+] T cells in the spleens of co-infected mice prompted us to test the capacity of these CD8[+] T cells to migrate to the brain during co-infection. We designed an *in vivo* migration assay where equal number of CD8[+] T cells isolated from the splenocytes of either single PbA-infected donors or co-infected donors at 6 dpi was adoptively transferred into single PbA-infected recipient mice at 5 dpi. Migration capacity of total LFA-1 or Pb1-specific CD8[+] T cells originating from the infected donors was quantified 22 h post-transfer by comparing the ratio of recovered infected donor cells in the brain to the numbers of cell initially transferred into the recipient. Interestingly, LFA-1[+] and Pb1-specific CD8[+] T cells originating from the co-infected donors migrated less efficiently to the brain than cells from single PbA-infected donors (Fig 5A).

To understand how co-infection alters the migration capacity of CD8[+] T cells toward the brain, splenic CD8[+] T cells were sorted and quantified by gene expression using NanoString (Geiss *et al*, 2008). Compared to single PbA-infected mice, *cd43, cd44, cd29, vla-4, lfa-1, cd62L, cxcr3, cxcr4, cxcr5, cxcr6, ccr5, ccr7,* and *ccr9* genes were differentially expressed in the co-infected mice (Appendix Fig S1A). We then assessed the surface expression of these gene products on parasite-specific CD8[+] T cells using flow cytometry (Appendix Fig S1B and C). The only differences observed between the splenic Pb1-specific CD8[+] T cells of single PbA-infected and co-infected mice were lower expression of CD43 and CXCR3 in the latter (Fig 5B and Appendix Fig S1C).

The possible roles of these two markers during co-infection were further investigated in detail. Although CD43 was previously shown to be important for T-cell trafficking to the brain during viral infection (Onami *et al*, 2002), the role of CD43 in ECM induction is unknown. Here, co-infection reduced the expression of CD43 in splenic CD8[+] T cells and decreased the number of CD43[+] CD8[+] T cells in the brain (Appendix Fig S2A and B). To decipher the role of CD43 in ECM, CD43[−/−] mice were infected with PbA. These mice developed ECM just like wild-type mice (Appendix Fig S2C and D), demonstrating that this molecule is dispensable for ECM. Hence, reduced expression of CD43 is unlikely to be the cause of ECM protection during concurrent co-infection.

## Co-infection does not affect migration of CD8[+] T cell to CXCR3 chemokines despite suppression of CXCR3

Unlike CD43, the importance of the chemokine receptor CXCR3 for CD8[+] T-cell migration to the brain during ECM has been clearly established (Campanella *et al*, 2008; Miu *et al*, 2008; Van den Steen *et al*, 2008). During co-infection, CD8[+] T cells had lower CXCR3 levels in the spleen, with the difference being especially stark in Pb1-specific CD8[+] T cells (Fig 5B and C). In addition, higher numbers of CXCR3[+]CD8[+] T cells were observed in the spleen of co-infected mice although the number of CXCR3[+] Pb1-specific CD8[+] T cells was similar (Fig 5D). To test whether this suppression of CXCR3 surface levels affects the ability of CD8[+] T cells to migrate in response to its cognate chemokines, Transwell migration assays were performed with isolated CD8[+] T cells from the spleen (Fig 5E

and F). Surprisingly, despite having a reduced CXCR3 surface expression, the LFA-1[+] (activated) and Pb1-specific CD8[+] T cells from co-infected mice were not impaired in their ability to migrate in response to CXCL9 and CXCL10 (Fig 5E and F). Given the lack of CXCL11 in C57BL/6J mice (Carr *et al*, 2008), CXCL11-dependent migration was not tested.

## Co-infection enhances splenic CXCL9 and CXCL10 levels to induce splenic retention of T cells

The reduced CXCR3 surface levels on CD8[+] T cells in co-infected mice could be the outcome and not the cause for the impaired spleen-to-brain migration. It has been previously shown that CXCR3 is internalized in response to high ligand levels (Colvin *et al*, 2004; Meiser *et al*, 2008). In addition, we have previously observed that during ECM, CXCR3 surface levels on CD8[+] T cells were lower in the brain than in the spleen (Poh *et al*, 2014), presumably because of higher CXCL9 and CXCL10 levels in the brain. We thus hypothesized that during CHIKV-PbA co-infection, higher levels of CXCR3-cognate chemokines may be induced in the spleen to mediate splenic retention of CD8[+] T cells and internalization of surface CXCR3. Indeed, significantly higher levels of CXCL9 and CXCL10 were found in the spleen in the co-infected mice at 6 dpi (Fig 6A and B). In contrast, similar levels of CXCL9 and CXCL10 levels were observed in the brain of single PbA-infected and co-infected mice at 6 dpi (Fig 6C).

To verify whether co-infection altered the chemotactic environment in the spleen and favored retention of T cells, an *in vivo* splenic retention assay was developed, where pooled CFSE-labeled splenocytes were transferred from single PbA-infected donors into either single PbA-infected or co-infected recipients at 5 dpi. Profiling of donor CD8[+] T cells retained in the recipients' spleen was done 22 h post-transfer. More donor CD8[+] T cells were found in the spleens of co-infected mice compared to single PbA-infected mice (Fig 6D–F). In particular, splenic retention of LFA-1[+] (activated) and Pb1-specific CD8[+] T cells in the co-infected recipients was significantly higher (> 10-folds) than in single PbA-infected recipients (Fig 6E and F).

To further demonstrate that the increased splenic levels of CXCR3-cognate chemokines mediated higher T-cell retention during co-infection, the same assay was performed using single PbA-infected CXCR3[−/−] donors. Although the retention of CD8[+] T cells was still higher in the co-infected recipients when single PbA-infected CXCR3[−/−] donor splenocytes were used (Fig 6D–F), the degree of increase in retention of LFA-1[+] (activated) and Pb1-specific CD8[+] T cells was significantly reduced to only ~2–3-folds as compared to a > 10-folds higher retention when WT donor splenocytes were used (Fig 6E and F and Appendix Fig S3). Interestingly, the retention of total and LFA-1[+] CD4[+] T cells was also higher in the co-infected recipients (~5-folds), but their retention was not dependent on CXCR3 (Appendix Fig S4).

## Co-infection induces high levels of splenic IFNγ to elevate CXCL9/CXCL10 and suppress CXCR3 expression to limit brain migratory capacity of CD8[+] T cells

The major inducer of CXCL9 and CXCL10 is type I interferon and IFNγ (Carter *et al*, 2007; Groom & Luster, 2011). However, at 6 dpi,

**Figure 5. Concurrent co-infection abrogates CD8+ T-cell migratory capacity to the brain and surface expression of CXCR3 in the spleen.**

A    *In vivo* migration assay measuring the migratory capacity of total, LFA-1+, and Pb1-specific CD8+ T cells from PbA donors (n = 5) and PbA + CHIKV donors (n = 10) toward the brain of PbA recipients. $5 \times 10^6$ isolated donors' CD8+ T cells (6 dpi) were transferred into PbA recipient at 5 dpi and harvested 22 h post-transfer. All data are expressed as ratio of recovered cells to initial numbers of cell transferred into the recipients for each specific cell type. Mann–Whitney two-tailed analysis (LFA-1+CD8+ T cells: *P = 0.0193, Pb1-specific CD8+ T cells: *P = 0.0280). Data shown were pooled from two independent experiments.

B    Representative histogram of CXCR3 expression in naïve CD8+ T cells, Pb1-specific CD8+ T cells from PbA mice, and Pb1-specific CD8+ T cells from co-infected mice is shown. Dotted line represents threshold for delineating CXCR3+ cells.

C    Surface expression of CXCR3 in total CD8+ T cells and Pb1-specific CD8+ T cells in the spleen of PbA (n = 6) and PbA + CHIKV (n = 7) on 6 dpi. Surface expression is determined by the geometric mean of CXCR3 signal by flow cytometry. Data shown are representative of two independent experiments. Mann–Whitney two-tailed analysis (total CD8+ T cells: **P = 0.0012, Pb1-specific CD8+ T cells: **P = 0.0012).

D    Number of CXCR3+CD8+ T cells and CXCR3+Pb1+CD8+ T cells in the spleen of PbA (n = 6) and PbA + CHIKV (n = 7) on 6 dpi. Data shown are a representative of two independent experiments. Mann–Whitney two-tailed analysis (Total CD8+ T cells: *P = 0.035).

E, F    Transwell migration assay using CXCL9 and CXCL10 with isolated total CD8+ cells from PbA (n = 6) and PbA + CHIKV (n = 6) on 6 dpi. Chemotaxis index was determined as the (cells across Transwell with CXCL9 or CXCL10/cells across Transwell without chemokines). Data shown are representative of two independent experiments. Mann–Whitney two-tailed analysis (for CXCL10 total CD8+ T cells: **P = 0.0022).

Data information: All data expressed as mean ± SD.

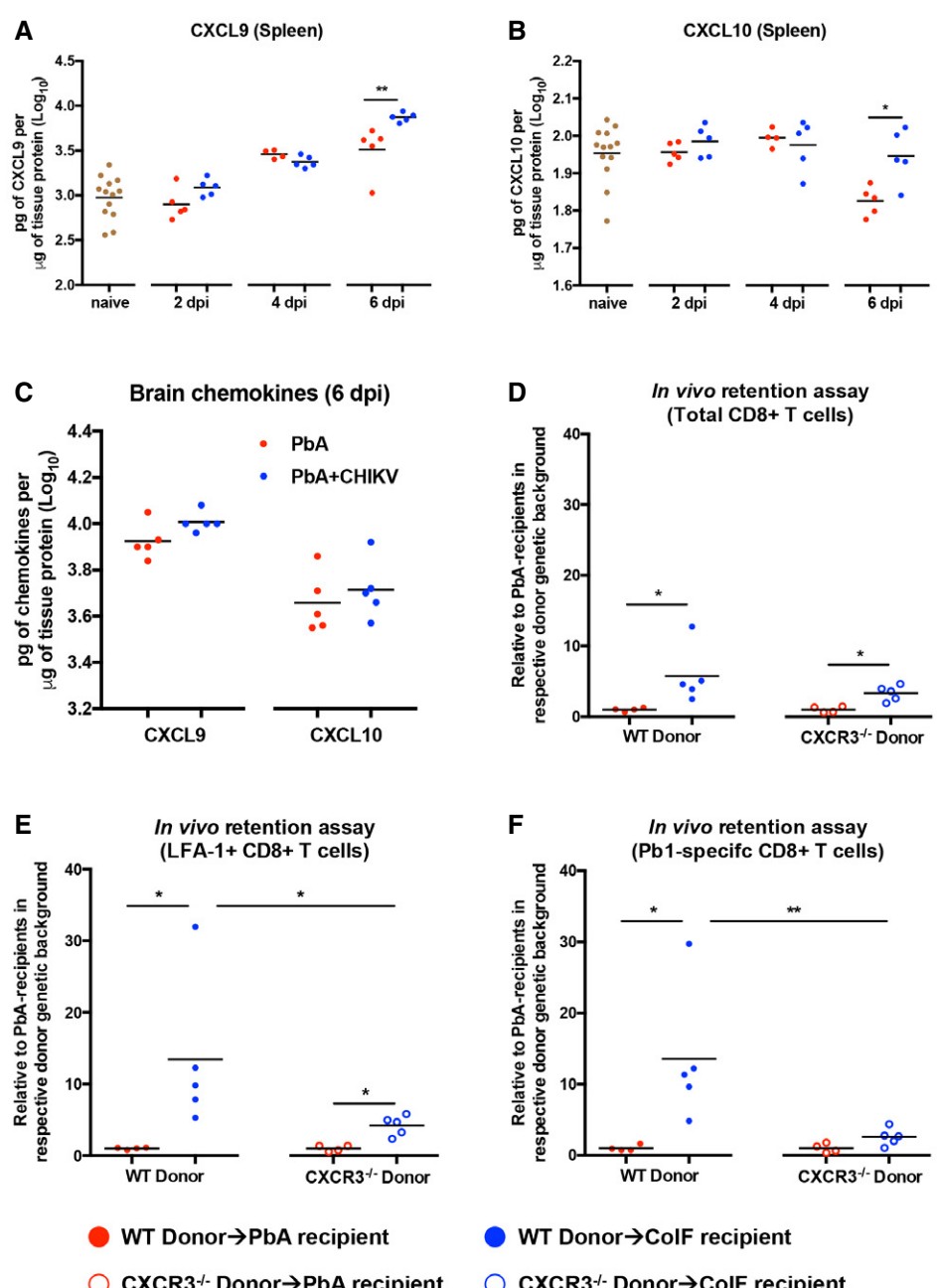

**Figure 6. Increased CXCL9/CXCL10 levels in the spleen of co-infected mice drive CXCR3-dependent splenic retention of CD8+ T cells.**

A    Levels of CXCL9 protein in the spleen of naïve, PbA, and PbA + CHIKV groups on 2, 4, and 6 dpi ($n \geq 5$ per group). Each data point was obtained from 1 mouse. Data comparison between PbA and PbA + CHIKV groups was done by Mann–Whitney two-tailed analysis (6 dpi; **$P$ = 0.0079). Data shown were pooled from three independent experiments.

B    Levels of CXCL10 protein in the spleen of naïve, PbA, and PbA + CHIKV groups on 2, 4, and 6 dpi ($n \geq 5$ per group). Each data point was obtained from 1 mouse. Data comparison between PbA and PbA + CHIKV groups was done by Mann–Whitney two-tailed analysis (6 dpi; *$P$ = 0.0317). Data shown were pooled from three independent experiments.

C    Levels of CXCL9 and CXCL10 level in the brain of PbA ($n$ = 5) and PbA + CHIKV ($n$ = 5) on 6 dpi.

D–F    *In vivo* retention assay displaying fold increase in recovered donors' cells relative to the mean of recovered cells in PbA recipients in the respective genetic backgrounds for total, LFA-1[+], and Pb1-specific CD8[+] T cells in each recipient spleen. WT Donor→PbA recipient ($n$ = 4), WT Donor→Co-infected (CoIF) recipient ($n$ = 5), CXCR3[−/−] donor→PbA recipient ($n$ = 4) and CXCR3[−/−] donor→Co-infected recipient ($n$ = 5). All data were analyzed by Mann–Whitney two-tailed analysis. For total CD8[+] T cells (D): WT donor→PbA recipient versus WT donor→CoIF recipient; *$P$ = 0.0159, CXCR3[−/−] donor→PbA recipient versus CXCR3[−/−] donor→CoIF recipient; *$P$ = 0.0159. For LFA-1[+]CD8[+] T cells (E): WT donor→PbA recipient versus WT donor→CoIF recipient; *$P$ = 0.0159, CXCR3[−/−] donor→PbA recipient versus CXCR3[−/−] donor→CoIF recipient; *$P$ = 0.0159, WT donor→CoIF recipient versus CXCR3[−/−] donor→CoIF recipient; *$P$ = 0.0159. For Pb1-specific CD8[+] T cells (F): WT donor→PbA recipient versus WT donor→CoIF recipient; *$P$ = 0.0159, WT donor→CoIF recipient versus CXCR3[−/−] donor→CoIF recipient; **$P$ = 0.0079.

Data information: Chemokine protein levels were determined as pg per μg of total protein, measured by ELISA using cell lysate from the spleen in (A and B) or brain in (C).

**Figure 7.   Elevated splenic IFNγ during co-infection induces splenic CXCL9/CXCL10 and suppresses CXCR3 expression on CD8$^+$ T cells to limit migration capacity toward the brain.**

A       Levels of IFNγ protein in the spleen of naïve, PbA, and PbA + CHIKV groups on 2, 4, and 6 dpi ($n \geq 5$ per group). Data comparison between PbA and PbA + CHIKV groups was done by Mann–Whitney two-tailed analysis (6 dpi; *$P$ = 0.0317).

B, C    Levels of CXCL9 and CXCL10 protein in the spleen of WT naïve ($n = 5$), WT + PbA ($n = 5$), WT + PbA + CHIKV ($n = 5$), IFNγ$^{-/-}$ + PbA ($n = 4$), and IFNγ$^{-/-}$ + PbA + CHIKV ($n = 4$) on 6 dpi. Data comparison between PbA and PbA + CHIKV groups in the respective WT and IFNγ$^{-/-}$ background was done by Mann–Whitney two-tailed analysis (CXCL9–WT + PbA versus WT + PbA + CHIKV; *$P$ = 0.0238, IFNγ$^{-/-}$ + PbA versus IFNγ$^{-/-}$ + PbA + CHIKV; $^{ns}P$ = 0.7429. CXCL10–WT + PbA versus WT + PbA + CHIKV; **$P$ = 0.0079, IFNγ$^{-/-}$ + PbA versus IFNγ$^{-/-}$ + PbA + CHIKV; $^{ns}P$ = 0.3429).

D       Number of total CD8$^+$ T cells and Pb1-specific CD8$^+$ T cells in the spleen of WT + PbA ($n = 5$), WT + PbA + CHIKV ($n = 4$), IFNγ$^{-/-}$ + PbA ($n = 5$), and IFNγ$^{-/-}$ + PbA + CHIKV ($n = 6$) on 6 dpi. Data comparison between PbA and PbA + CHIKV groups in the respective WT and IFNγ$^{-/-}$ background was done by Mann–Whitney two-tailed analysis (total CD8$^+$ T cells–WT + PbA versus WT + PbA + CHIKV; *$P$ = 0.0159, IFNγ$^{-/-}$ + PbA versus IFNγ$^{-/-}$ + PbA + CHIKV; $^{ns}P$ = 0.0823, Pb1-specific CD8$^+$ T cells–WT + PbA versus WT + PbA + CHIKV; *$P$ = 0.0317, IFNγ$^{-/-}$ + PbA versus IFNγ$^{-/-}$ + PbA + CHIKV; $^{ns}P$ = 0.4286).

E, F    CXCR3 surface expression on total CD8$^+$ T cells and Pb1-specific CD8$^+$ T cells in the spleen of WT + PbA ($n = 5$), WT + PbA + CHIKV ($n = 4$), IFNγ$^{-/-}$ + PbA ($n = 5$), and IFNγ$^{-/-}$ + PbA + CHIKV ($n = 6$) on 6 dpi. Representative histograms showing CXCR3 expression are shown. Threshold of CXCR3$^+$ cells is delineated by black dotted line. Data comparison between PbA and PbA + CHIKV groups in the respective WT and IFNγ$^{-/-}$ background was done by Mann–Whitney two-tailed analysis (total CD8$^+$ T cells–WT + PbA versus WT + PbA + CHIKV; *$P$ = 0.0159, IFNγ$^{-/-}$ + PbA versus IFNγ$^{-/-}$ + PbA + CHIKV; $^{ns}P$ = 0.0823, Pb1-specific CD8$^+$ T cells–WT + PbA versus WT + PbA + CHIKV; *$P$ = 0.0159, IFNγ$^{-/-}$ + PbA versus IFNγ$^{-/-}$ + PbA + CHIKV; $^{ns}P$ = 0.9307).

G       *In vivo* migration assay measuring the migratory capacity of LFA-1$^+$ and Pb1-specific CD8$^+$ T cells from IFNγ$^{-/-}$ + PbA donors ($n = 4$) and IFNγ$^{-/-}$ + PbA + CHIKV donors ($n = 3$) toward the brain of WT PbA recipients. $7 \times 10^6$ isolated donors' CD8$^+$ T cells (6 dpi) were transferred into PbA recipient at 5 dpi and harvested 22 h post-transfer. All data are expressed as ratio of recovered cells to initial numbers of cell transferred into the recipients for each specific cell type. Mann–Whitney two-tailed analysis (LFA-1$^+$CD8$^+$ T cells: $^{ns}P$ = 0.9999, Pb1-specific CD8$^+$ T cells: $^{ns}P$ = 0.6286).

Data information: For all cytokines or chemokines, quantifications were measured by ELISA using cell lysate from the organ and determined as pg/µg of total protein. Each data point shown in the dot plots was obtained from 1 mouse.

the time point where CXCL9 and CXCL10 are elevated in co-infected mice, type I interferon is suppressed in CHIKV-infected mice while IFNγ is highly induced (Teo *et al*, 2015). In addition, we showed that IFNγ was induced in the spleen of single-CHIKV-infected mice at 6 dpi (Appendix Fig S5). As such, we investigated whether higher levels of these chemokines in the spleens of co-infected mice could be attributed to higher IFNγ levels locally induced in the spleen. ELISA results from splenocytes lysate showed that more IFNγ was present in the spleens of co-infected mice at 6 dpi compared to singly PbA-infected mice (Fig 7A).

To further demonstrate the causal relationship between IFNγ and CXCL9/CXCL10 induction in the co-infected mice, we performed co-infection in IFNγ$^{-/-}$ mice. Similar to previous study, ECM does not occur in IFNγ$^{-/-}$ mice (Claser *et al*, 2011); hence, ECM was not recapitulated in the co-infected IFNγ$^{-/-}$ mice (Appendix Fig S6). However, induction of splenic CXCL9 (~10-fold reduction) and CXCL10 (~4-fold reduction) was abolished in co-infected IFNγ$^{-/-}$ mice (Fig 7B and C) and restored the numbers of splenic total and Pb1-specific CD8$^+$ T cells. CXCR3 expression in the co-infected IFNγ$^{-/-}$ mice was similar to single PbA-infected mice (Fig 7D–F). Importantly, when isolated CD8$^+$ T cells (6 dpi) from single PbA-infected or co-infected donors in IFNγ$^{-/-}$ background were transferred into WT PbA-infected recipients, a similar proportion of LFA-1$^+$ and Pb1-specific CD8$^+$ T cells was recovered in the recipient brains (Fig 7G), suggesting a restoration of migration capacity of these CD8$^+$ T cells in the IFNγ$^{-/-}$ co-infected donor.

**Splenic CD4$^+$ T cells is the major contributor of IFNγ during co-infection**

To identify leukocyte subsets responsible for the higher levels of splenic IFNγ during co-infection, intracellular staining for IFNγ was performed on splenocytes harvested on 6 dpi. CD4$^+$ T cells, CD8$^+$ T cells, NK cells, NKT cells, and neutrophils were identified to be the main IFNγ producers in the spleen (Fig 8A). Interestingly, only CD4$^+$ T cells showed the most prominent induction of IFNγ

expression and increase in IFNγ$^+$ numbers upon co-infection (Fig 8). Co-infection did not alter IFNγ expression in CD8$^+$ T cells. Total and IFNγ-producing NK cells were reduced with parasite infection. IFNγ-producing NKT cells were a minor producing subset in comparison with other subsets (Fig 8); hence, they are unlikely to be the major contributors to this phenomenon. For neutrophils, while IFNγ expression is high in all four groups, the difference between single PbA infection and co-infection was a result of neutrophils reduction in single PbA infection and not an increase in IFNγ expression level in the co-infected mice (Fig 8). Taken together, CD4$^+$ T cells are the major contributors of higher splenic IFNγ during co-infection.

# Discussion

Worms [*Schistosoma sp* (Hartgers & Yazdanbakhsh, 2006; Waknine-Grinberg *et al*, 2010; Bucher *et al*, 2011; Wang *et al*, 2013, 2014), *Litomosoides sigmodontis* (Karadjian *et al*, 2014), and *Heligmosomoides polygyrus* (Su *et al*, 2005)] and bacteria [*Mycobacterium tuberculosis* (Li & Zhou, 2013), *Listeria monocytogenes* (Qi *et al*, 2013), and *Salmonella* (Cunnington *et al*, 2012)] have been shown to alter the pathological outcome of malaria infection through bystander regulation of the host immunity. Specific to the PbA-ECM model, the mechanisms leading to ECM protection during co-infection were mostly due to the suppression of the pro-inflammatory response induced by the heterologous pathogen, which occurs from 4 days post-*Plasmodium* infection onwards. In helminth–*Plasmodium* co-infection, *S. japonicum* (Wang *et al*, 2013, 2014) and *S. mansoni* (Waknine-Grinberg *et al*, 2010; Bucher *et al*, 2011) protected co-infected mice from ECM by inducing a Th2-polarizing response in the host, shifting the balance away from the Th1 response that is essential for ECM to occur. In nematode–*Plasmodium* co-infection, *L. sigmodontis* induced an anti-inflammatory IL-10 response in the co-infected host, which counteracted the Th1 response and protected the host from ECM (Specht *et al*, 2010).

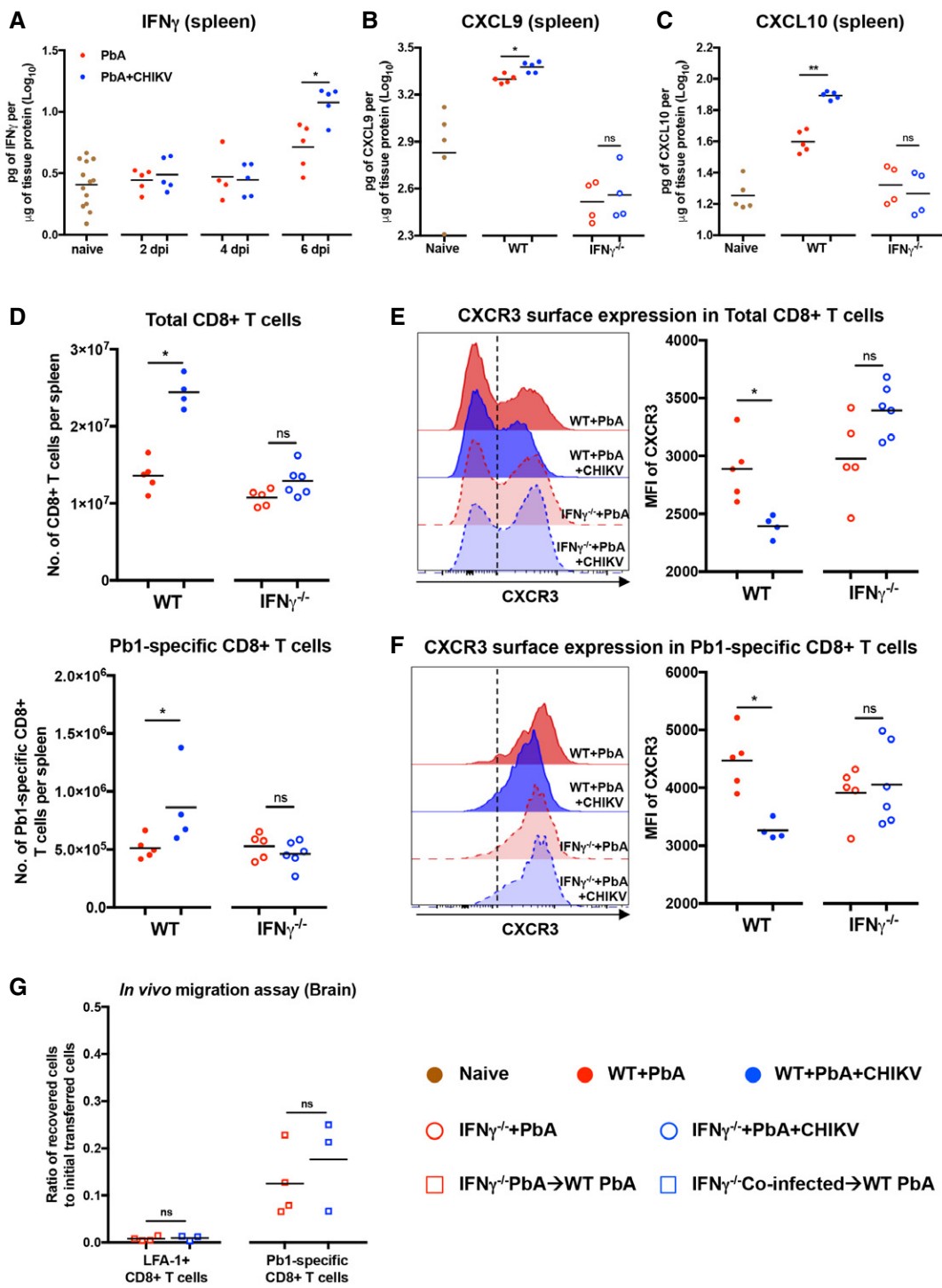

**Figure 7.**

Similarly, the murine AIDS virus was reported to protect against ECM by inducing splenic CD4 T cells to express IL-10 (Eckwalanga *et al*, 1994).

We show in this study that concurrent co-infection with CHIKV also protected the host from ECM. A novel interfering mechanism was identified whereby co-infection with CHIKV alters the trafficking of pathogenic parasite-specific CD8$^+$ T cells (Fig 8). In

particular, concurrent co-infection with CHIKV abolishes early migration of CD8$^+$ T cells to the brain. Instead, these cells are retained in the spleen, thus protecting co-infected mice from ECM-induced neuropathology. Co-infection induces higher IFNγ levels in the spleen compared to single PbA infection. The higher IFNγ is likely generated by IFNγ-producing CHIKV-specific CD4$^+$ T cells in the spleen which was previously shown to be elevated on 6 days

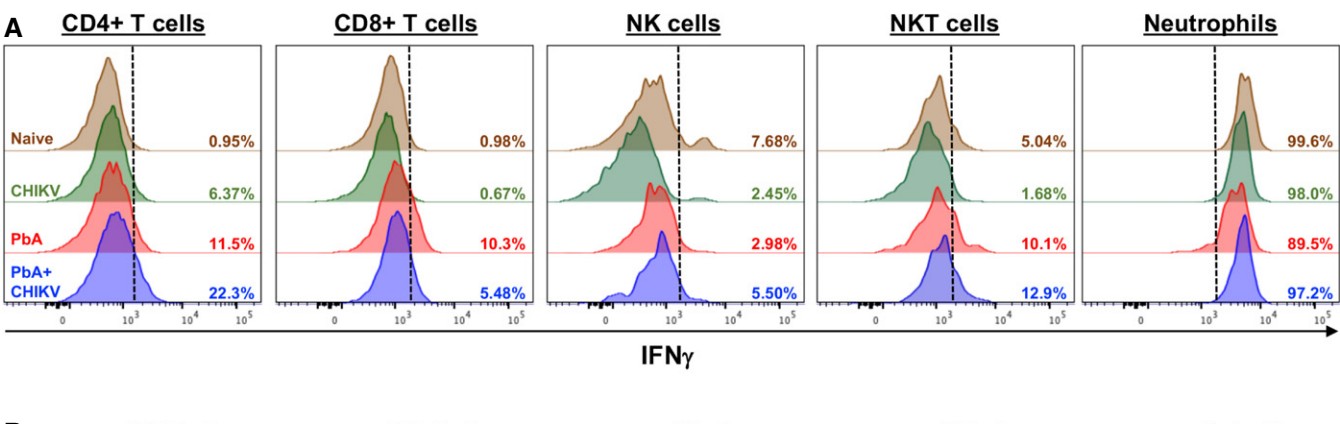

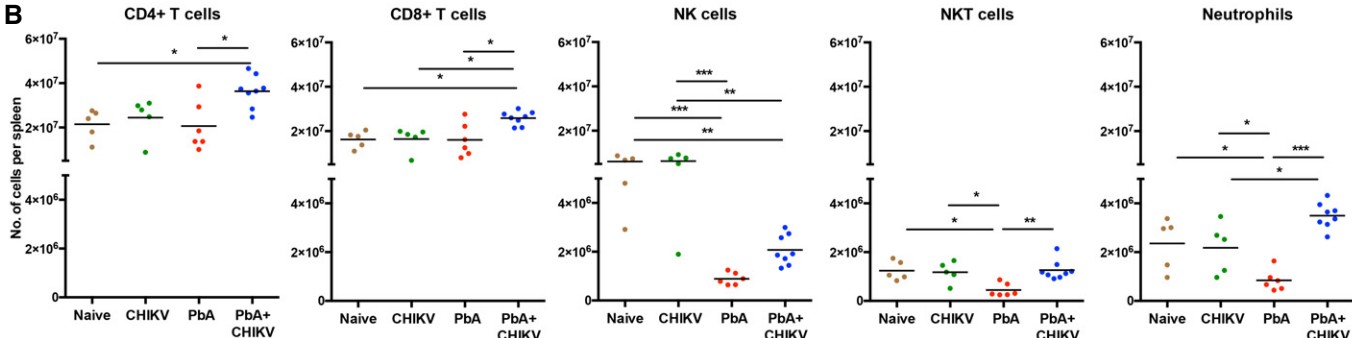

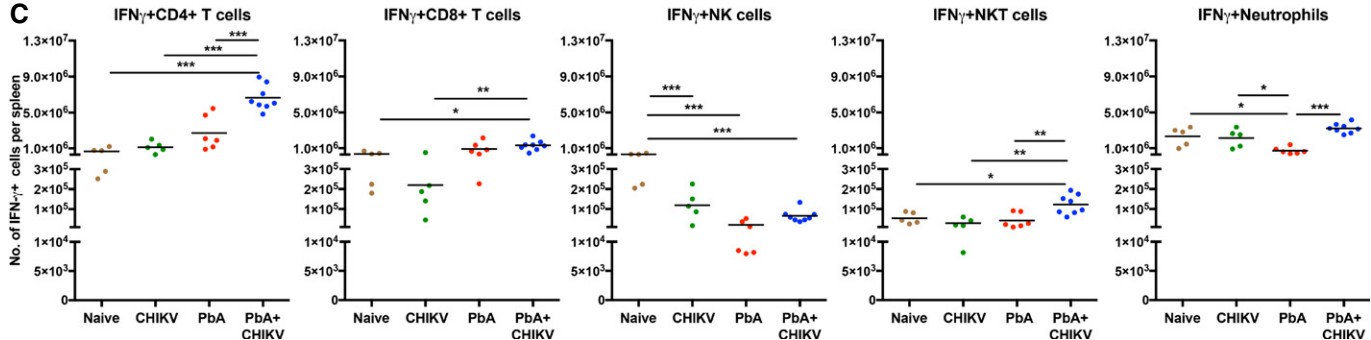

**Figure 8.  Increased IFNγ producing CD4⁺ T cells in the spleen drives the enhanced splenic IFNγ on 6 dpi during concurrent co-infection.**

A   Representative histogram showing IFNγ production in CD4⁺ T cells, CD8⁺ T cells, NK cells, NKT cells, and neutrophils in the spleen of naïve ($n = 5$), CHIKV ($n = 5$), PbA ($n = 6$), and PbA + CHIKV ($n = 8$) on 6 dpi. Black dotted line represents threshold setting for IFNγ⁺ cells.

B   Numbers of CD4⁺ T cells, CD8⁺ T cells, NK cells, NKT cells, and neutrophils in the spleen of naïve ($n = 5$), CHIKV ($n = 5$), PbA ($n = 6$), and PbA + CHIKV ($n = 8$) on 6 dpi. CD4⁺, CD8⁺ T cells, NK cells, NKT cells, and neutrophils were defined as CD3⁺CD4⁺, CD3⁺CD8⁺, CD3⁻NK1.1⁺, CD3⁺NK1.1⁺, and CD3⁻CD11b⁺Ly6G⁺ cells, respectively. All data analyzed by one-way ANOVA with Tukey's post-test. For CD4⁺ T cells: naïve versus PbA + CHIKV; *mean diff* = $-1.49 \times 10^7$, PbA versus PbA + CHIKV; *mean diff* = $-1.57 \times 10^7$. For CD8⁺ T cells: naïve versus PbA + CHIKV; *mean diff* = $-9.61 \times 10^6$, CHIKV versus PbA + CHIKV; *mean diff* = $-9.43 \times 10^6$, PbA versus PbA + CHIKV; *mean diff* = $-9.75 \times 10^6$. For NK cells: naïve versus PbA; ***mean diff* = $5.22 \times 10^6$, naïve versus PbA + CHIKV; **mean diff* = $4.03 \times 10^6$, CHIKV versus PbA; ***mean diff* = $5.39 \times 10^6$, CHIKV versus PbA + CHIKV; **mean diff* = $4.20 \times 10^6$. For NKT cells: naïve versus PbA; *mean diff* = $7.95 \times 10^5$, CHIKV versus PbA; *mean diff* = $7.34 \times 10^5$, PbA versus PbA + CHIKV; **mean diff* = $-8.19 \times 10^5$. For neutrophils: naïve versus PbA; *mean diff* = $1.51 \times 10^6$, CHIKV versus PbA; *mean diff* = $1.34 \times 10^6$, CHIKV versus PbA + CHIKV; *mean diff* = $-1.32 \times 10^6$, PbA versus PbA + CHIKV; ***mean diff* = $-2.66 \times 10^6$.

C   Numbers of IFNγ-producing CD4⁺ T cells, CD8⁺ T cells, NK cells, NKT cells, and neutrophils in the spleen of naïve ($n = 5$), CHIKV ($n = 5$), PbA ($n = 6$) and PbA + CHIKV ($n = 8$) on 6 dpi. All data analyzed by one-way ANOVA with Tukey's post-test. For IFNγ⁺CD4⁺ T cells: naïve versus PbA + CHIKV; ***mean diff* = $-5.99 \times 10^6$, CHIKV versus PbA + CHIKV; ***mean diff* = $-5.52 \times 10^6$, PbA versus PbA + CHIKV; ***mean diff* = $-3.95 \times 10^6$. For IFNγ⁺CD8⁺ T cells: naïve versus PbA + CHIKV; *mean diff* = $-9.28 \times 10^5$, CHIKV versus PbA + CHIKV; **mean diff* = $-1.08 \times 10^6$. For IFNγ⁺ NK cells: naïve versus CHIKV; ***mean diff* = $1.83 \times 10^5$, naïve versus PbA; ***mean diff* = $2.81 \times 10^5$, naïve versus PbA + CHIKV; ***mean diff* = $2.83 \times 10^5$. For IFNγ⁺ NKT cells: naïve versus PbA + CHIKV; *mean diff* = $-68295$, CHIKV versus PbA + CHIKV; **mean diff* = $-93362$, PbA versus PbA + CHIKV; **mean diff* = $-80084$. For IFNγ⁺ neutrophils: naïve versus PbA; *mean diff* = $1.58 \times 10^6$, CHIKV versus PbA; *mean diff* = $1.40 \times 10^6$, PbA versus PbA + CHIKV; ***mean diff* = $-2.50 \times 10^6$.

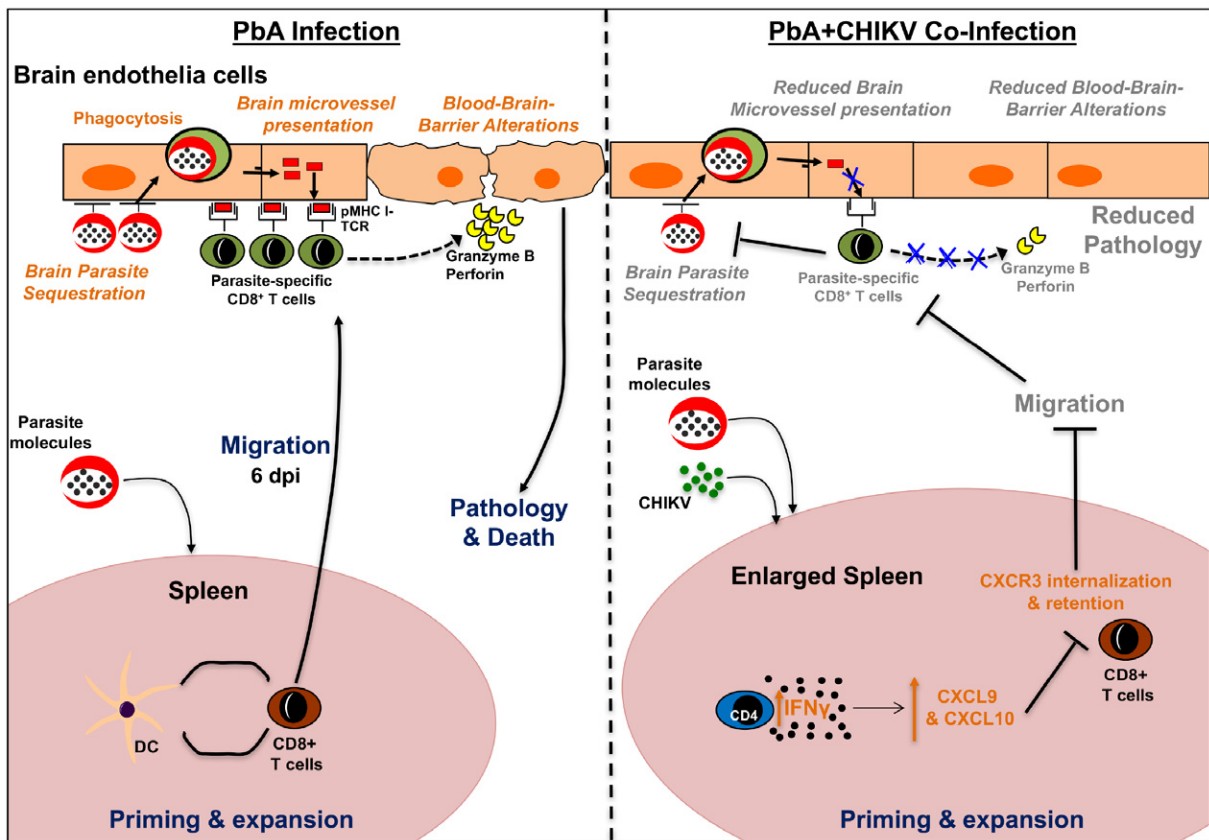

**Figure 9.  Proposed mechanism of ECM protection during concurrent co-infection of PbA and CHIKV.**

In normal PbA infection (shown on the left) in susceptible C57BL/6J, parasite-specific pathogenic CD8[+] T cells are generated in the spleen. These CD8[+] T cells migrate to the brain and interact with brain endothelia cross-presenting parasite proteins, leading to cytotoxic events that alter the BBB, causing eventual death. Pathogenic events in the brain are highlighted in italics brown font. In concurrent co-infection with PbA and CHIKV (shown on the right), increased IFNγ production is induced in splenic CD4[+] T cells. This drives the enhanced production of CXCL9 and CXCL10 in the spleen. Increased CXCR3–chemokine interaction leads to internalization and retention of CXCR3 on the pathogenic CD8[+] T cells in the spleen. This reduces migration of these cells into the brain and abrogates downstream pathogenic events mediated by parasite-specific CD8[+] cells. Pathogenic events suppressed during co-infection are highlighted in gray font.

post-CHIKV infection (Teo *et al*, 2017). We propose that the higher levels of splenic IFNγ in these co-infected mice drive higher local production of CXCL9 and CXCL10 (Carter *et al*, 2007). This promotes splenic retention of pathogenic CD8[+] T cells induced in the spleen and reduces trafficking of these cells to the brain. Furthermore, in line with a previous study showing that CD8[+] T cells regulate parasite sequestration in the brain (Claser *et al*, 2011), co-infected mice have lower parasite loads in the brain and hence diminish brain endothelial cross-presentation of parasite antigens. With lesser parasite-specific CD8[+] T cells and fewer cross-presenting targets of endothelial cells, BBB integrity is preserved, thereby protecting mice from ECM.

While the drastic differences between co-infected and PbA-infected mice in the splenic retention assay suggest that chemokine milieu in the spleen is primarily responsible for the altered T-cell migration patterns, there remains a minor role for differences in the migration capacity of the CD8[+] T cells themselves. A modest decrease in the ability of adoptively transferred CD8[+] T cells from co-infected donors to migrate to the brains of PbA-infected recipients was observed as compared to single PbA-infected donors. Extensive phenotyping found two molecules that could have a role

behind this reduced migration capacity *in vivo*. Of the two molecules found to be differentially expressed in the parasite-specific CD8[+] T cells of co-infected, CD43 was found to be dispensable for ECM, whereas the reduced surface expression levels of CXCR3 did not affect the ability of these cells to migrate *in vitro* in response to CXCL9 and CXCL10. Instead, the low level of surface CXCR3 expression on the splenic CD8[+] T cells of co-infected mice is likely to be a consequence of high exposure to CXCL9 and CXCL10 in the spleen, leading to receptor internalization and degradation (Colvin *et al*, 2004; Meiser *et al*, 2008). In addition to CXCL9 and CXCL10, other chemokines may also be highly induced by IFNγ in the spleens of co-infected mice. This can be deduced by the results of the *in vivo* splenic retention assay using donor cells deficient in CXCR3, where the enhanced ability of the co-infected mouse spleens to attract the CD8[+] T cells was largely, but not completely abrogated.

While co-infection with CHIKV is protective in the context of ECM, co-infection of CHIKV with other pathogens could be detrimental if CD8[+] T cells are needed for protection in the peripheral tissues. In contrast to their pathogenicity in ECM, CD8[+] T cells are needed for protection against a wide spectrum of neurological

infections including West Nile virus (Klein *et al*, 2005; Brien *et al*, 2007), human simplex virus-1 (Lang & Nikolich-Zugich, 2005), cytomegalovirus (Bantug *et al*, 2008), enterovirus 71 (Lin *et al*, 2009), *Toxoplasma gondii* (Guiton *et al*, 2009), neuroborreliosis (Jacobsen *et al*, 2003), and central nervous system listeriosis (Schluter *et al*, 1995; Hayashi *et al*, 2009). The IFN$\gamma$-dependent mechanism described here induced by co-infection with CHIKV could possibility alter or delay CD8$^+$ T-cell migration to the brain, leading to disease aggravation. The ECM results here underscore the importance of further studies into the regulation and function of T-cell migration in the context of CHIKV co-infection with other pathogens. Such co-infections may interfere with protective or pathogenic immune responses induced by single infection. Understanding the interfering mechanisms will pave the way in the development of new therapeutic modalities. Lastly, it also calls for comprehensive pathogen identification in endemic countries when symptomatic diagnosis of one pathogen may be confounded by another pathogen.

# Materials and Methods

### Mice

Six-week-old male or female CD45.2, IFN$\gamma^{-/-}$, CXCR3$^{-/-}$, and CD43$^{-/-}$ mice in C57BL/6J background were used. CD43$^{-/-}$ mice in B6;129 background was purchased from Jackson Lab and backcrossed for three generations with C57BL/6J to obtain CD43$^{-/-}$ F3 generation before use in any experiments. All mice were bred and kept under specific pathogen-free conditions in the Biological Resource Centre, Agency for Science, Technology and Research, Singapore. All experiments and procedures were approved by the Institutional Animal Care and Use Committee (IACUC: 140968) of the Agency for Science, Technology and Research, Singapore, in accordance with the guidelines of the Agri-Food and Veterinary Authority and the National Advisory Committee for Laboratory Animal Research of Singapore.

### Pathogens and infection

The transgenic *P. berghei* ANKA (231c1 l) line expressing luciferase and green fluorescent proteins under control of the ef1-a (PbA-luc) used in this study was kindly provided by Dr. Christian Engwerda (Amante *et al*, 2007). Infected RBC (iRBC) was prepared through *in vivo* passage in C57BL/6J mice and stored in Alsever's solution in liquid nitrogen (Claser *et al*, 2011).

The CHIKV strain used was isolated at the National University Hospital from the 2008 outbreak in Singapore (Her *et al*, 2010). CHIKV was propagated in C6/36 cultures and quantified by standard plaque assay in Vero E6 cells (Teo *et al*, 2013). Mice were infected with PbA-luc by inoculating 10$^6$ iRBC intraperitoneally (i.p.). CHIKV was inoculated subcutaneously in the ventral side of the right hind footpad toward the ankle with $1 \times 10^6$ PFU CHIKV in 30 µl PBS.

### Infection parameters and ECM evaluation

Parasitemia was monitored by flow cytometry daily from 3 to 12 dpi and subsequently on alternate days as previously described

(Malleret *et al*, 2011). Parasite accumulation in the whole body and head was also measured daily using an *in vivo* bioluminescence imaging system (IVIS Spectrum, Xenogen, Alameda, CA) as previously reported (Claser *et al*, 2011). Mice were considered to have ECM if they display neurological symptoms such as paralysis, ataxia, deviation of the head, convulsion, coma, and died within the ECM window of 6–12 dpi (Claser *et al*, 2011).

### Determination of parasite sequestration in the brain

Mice were injected with 200 µl of D-luciferin potassium salt (Caliper Life sciences) in PBS (5 mg/ml). After 2 min, mice were perfused, and individual brains were removed and placed on petri dish. Bioluminescence signals were quantified by *in vivo* bioluminescence imaging system (IVIS Spectrum, Xenogen, Alameda, CA) at field of vision (FOV) C (13.1 cm; Claser *et al*, 2011).

### Determination of BBB permeability

Evans blue permeability assay was used to quantify BBB permeability. Briefly, each mouse was given 200 µl of 1% w/v Evans blue (Sigma-Aldrich) in 0.9% w/v NaCl intravenously (i.v). After 1-h incubation, the mouse was anesthetized and perfused, and the brain was removed and placed in 1 ml of *N,N*-dimethylformamide (Sigma-Aldrich). Samples were incubated at room temperature for 48 h, and absorbance readings of the supernatant were taken at OD$_{620\ nm}$.

### *In vivo* cytotoxic assay

*In vivo* cytolysis assay was conducted to test the cytotoxic capacity of Pb1-specific (dominant CD8 epitope for PbA infection) CD8$^+$ T cells (Howland *et al*, 2013). Briefly, naive splenocytes were divided into two portions. One portion was incubated with 10 µg/ml SQLLNAKYL peptide for 1 h at 37°C and then washed and labeled with 0.5 µM carboxy-fluorescein succinimidyl ester (CFSE) for 10 min at 37°C; the other portion was not pulsed with peptide and labeled with 5 µM CFSE. Equal numbers of peptide-pulsed and unpulsed splenocytes (10$^7$ cells each) were injected (i.v) into naive mice, PbA-infected or PbA-CHIKV co-infected mice at 6 dpi. The mice were sacrificed 20 h later to analyze the CFSE-labeled cells in the spleen by flow cytometry.

### Leukocytes profiling in the brain and spleen

Mice were sacrificed and perfused by intracardial injection of PBS at 6 dpi. Subsequently, the brain and spleen were extracted and processed to obtain leukocytes as previously described (Claser *et al*, 2011; Teo *et al*, 2013). Isolated leukocytes were stained with LIVE/DEAD Aqua (Life Technologies), then blocked in 100 µl of blocking buffer consisting of a mix of 1% of rat and mouse serum (Sigma-Aldrich) in FACS buffer [1% BSA, 2 µm EDTA in PBS]. Next, cells were incubated with PE-labeled SQLLNAKYL-H-2D$^b$ (Pb-1) tetramer (Howland *et al*, 2013) on ices before addition of conjugated antibodies for another 20 min of incubation. Cells were fixed in IC fixation buffer (ebioscience) for 5 min before acquisition using a LSR II flow cytometer (BD Biosciences). Conjugated antibodies used were as follows: α-CD45 (clone 30-F11, BD Biosciences, 1:400 dilution),

α-CD3 (clone 17A2, BD Biosciences, 1:200 dilution), α-CD4 (clone GK1.5, Biolegend, 1:400 dilution), α-CD8 (clone 53–6.7, BD Biosciences, 1:400 dilution), α-LFA-1 (H155-78; Biolegend, 1:200 dilution), α-NK1.1 (clone PK136, ebioscience, 1:200 dilution), α-CD11b (clone M1/70, Biolegend, 1:400 dilution), and α-Ly6G (clone 1A8, Biolegend, 1:400 dilution). Gating strategy of T-cell infiltrate in the brain is shown in Appendix Fig S7. Majority of the T-cell infiltrate in the brain of the infected mice express LFA-1 marker (Appendix Fig S7).

### Brain microvessel cross-presentation assay

Brain microvessel cross-presentation assay for the Pb1 epitope of PbA was performed as described previously at 6 dpi (Howland *et al*, 2013, 2015a). Briefly, mice were perfused and the brain (without the meninges and brain stem) was finely minced with 1 ml of medium and homogenized by passing five times through a 23-gauge needle. The homogenate was mixed with an equal volume of 30% dextran (MW ~70,000, Sigma-Aldrich) in PBS and centrifuged at 10,000 $g$ for 15 min at 48°C. To retain the microvessels, the pellet was resuspended in PBS and passed through a 40-mm cell strainer. After washing to remove leukocytes, the cell strainer was backflushed with 2 ml PBS over a 6-well plate to collect the microvessels. Microvessels were rocked at room temperature with 2% FBS, 1 mg/ml of type 4 collagenase, and 10 mg/ml of DNase I for 90 min. The digested microvessels were added to 5 ml medium, pelleted at 500 $g$ for 5 min, resuspended in 500 ml of medium, and divided between five wells of a 96-well filter plate. LR-BSL8.4a reporter cells (3,104 cells in 100 ml) were added to each well before the plate was incubated overnight and then stained with X-gal. Degree of cross-presentation was quantified as blue spots using "ImmunoSpot 5.0 Analyzer Professional DC" (Cellular Technology Ltd).

### *In vivo* migration assay

At 6 dpi, spleens from singly PbA-infected or co-infected donors were harvested. Total CD8$^+$ T cells were isolated from these donors through negative selection with a CD8α$^+$ T-cell isolation kit (Miltenyi Biotec). Cells were subsequently labeled with 5 μM of CFSE solution for 10 min, washed with PBS. A solution containing $5 \times 10^6$ cells in 200 μl of PBS was injected i.v. into a PbA-infected recipient at 5 dpi. A portion of these donor cells were stained with α-CD45 (1:400), α-CD3 (1:200), α-CD4 (1:400), α-CD8 (1:400), α-LFA-1 (1:200) antibodies and Pb1 tetramer followed by flow cytometry acquisition to profile for the numbers of total, LFA-1$^+$ and Pb1-specific CD8$^+$ T cells in the $5 \times 10^6$ donor cells. Twenty-two hours post-transfer, brains were extracted from the recipient mice and processed to obtain cell suspension. Extracted cells were stained with live/dead determination dye (Invitrogen) for 20 min, followed by α-CD45 (1:400), α-CD3 (1:200), α-CD4 (1:400), α-CD8 (1:400), α-LFA-1 (1:200) antibodies, and Pb1 tetramer and flow cytometry acquisition using a LSR II flow cytometer. Migration capacity of the donor cells toward the brain was determined by the ratio of donor cells obtained from the recipient brain divided by initial number of donors cells transferred. Schematic diagram of this assay is provided in Appendix Fig S8.

### Transwell chemotaxis assay for CD8$^+$ T cells

On 6 dpi, splenocytes were isolated and CD8$^+$ T cells were selected through negative selection with a CD8a$^+$ T cells isolation kit (Miltenyi Biotec). 5-μm polyester membrane Transwell inserts (Corning) were placed on a 24-well plate (Corning) containing 600 μl of RPMI medium with 10% FBS and either no chemokines, 200 ng/ml of CXCL9, or 200 ng/ml of CXCL10. $1 \times 10^6$ isolated CD8$^+$ T cells were added to the upper chamber in 100 μl of medium lacking chemokines. The Transwell setup was incubated at 37°C for 3 h and activated CD3$^+$CD8$^+$LFA-1$^+$ or parasite-specific CD3$^+$CD8$^+$Pb1$^+$ T cells that crossed the membrane were quantified by flow cytometry. The chemotaxis index of each CD8$^+$ T-cell subset was determined by the ratio of cells recovered with chemokine-containing medium to cells recovered with medium alone.

### *In vivo* retention assay

At 6 dpi, splenocytes were isolated from PbA-infected donors, pooled at $5 \times 10^7$ cells/ml and labeled in 5 μM CFSE solution for 10 min. After labeling and washing, $5 \times 10^7$ donor splenocytes in 200 μl PBS was injected i.v. into each PbA or PbA + CHIKV recipient at 5 dpi. 22 h post-transfer, recipients' splenocytes were harvested and stained with α-CD45 (1:400), α-CD3 (1:200), α-CD4 (1:400), α-CD8 (1:400), α-LFA-1 (1:200), and Pb1-specific tetramer.

### Gene expression of splenic CD8$^+$ T cells by NanoString

On 6 dpi, splenocytes were isolated and CD8$^+$ T cells (CD3$^+$CD8$^+$ cells) were sorted (4-laser FACSAria III, BD Biosciences). Sorted cells were immediately lysed in lysis buffer from RNeasy kit (Qiagen) and kept at $-80$°C. Gene expression in 10,000–15,000 lysed cells was quantified using the nCounter$^{®}$ Mouse immunology kit (NanoString Technologies) following the manufacturer's protocols. Data were analyzed and visualized by TIBCO Spotfire$^{®}$ software (PerkinElmer).

### Surface expression of chemokine receptors

Isolated splenocytes at 6 dpi were stained with LIVE/DEAD Aqua followed by Pb-1-specific tetramer and surface markers staining. The following antibodies were used: α-CD62L (clone MEL-14, Biolegend, 1:200 dilution), α-CXCR4 (clone 2B11, ebioscience, 1:200 dilution), α-CD43 (clone 1B11, Biolegend, 1:200 dilution), α-CD44 (clone 1M7, BD Biosciences, 1:200 dilution), α-CCR5 (clone HM-CCR5, Biolegend, 1:200 dilution), α-CXCR3 (clone CXCR3-173, ebioscience, 1:200 dilution), α-LFA-1 (H155-78; Biolegend, 1:200 dilution), and VLA-4 (clone R1-2, Biolegend, 1:200 dilution). Levels of surface expression were determined by mean fluorescence intensity (MFI).

### Quantification of cytokine/chemokine proteins in cell lysate by ELISA

Spleen and brain were extracted and homogenized in RIPA buffer (50 mM Tris–HCl pH 7.4; 1% NP-40; 0.25% sodium deoxycholate; 150 mM NaCl; 1 mM EDTA) with 1× protease inhibitors (Roche) using a micro-bead cell disrupter (Micro Smash MS-100, Digital Biology) at 5,000 rpm for 1 min. Cell lysates were further sonicated at

**The paper explained**

**Problem**

The global incidences of Chikungunya virus (CHIKV) has risen significantly over the last decade, increasing the likelihood of co-infection with endemic malaria that shares similar geographical distribution. However, nothing is known about the impact CHIKV has on malaria during co-infection.

**Results**

Using the *P. berghei* ANKA (PbA) experimental cerebral malaria (ECM) model, we show that among the different co-infection scenarios, concurrent co-infection induced the most prominent changes in ECM manifestation. Concurrent co-infection protected mice from ECM-induced neuropathology by limiting trafficking of disease-causing CD8[+] T cells to the brain. This occurs through the induction of higher splenic IFNγ during co-infection, leading to higher local levels of CXCL9 and CXCL10 that retains the CXCR3-expressing CD8[+] T cells in the spleen. The absence of these CD8[+] T cells in the brain averts all downstream pathogenic events such as parasite sequestration in the brain and disruption of blood–brain barrier that prevents ECM-induced mortality in co-infected mice.

**Impact**

This is the first report demonstrating how infection disrupts the chemokine milieu and alters the pathological outcome of an unrelated infection. This new mechanism of immune interference could be highly relevant to a broad spectrum of pathogens during co-infection with arboviruses. Importantly, similar regulation of CD8[+] T cells could be detrimental and not protective during co-infection of CHIKV with other diseases.

70% intensity for 15 s (Branson Ultrasonics Sonifier™ S-450), and supernatants were collected to quantify IFNγ, CXCL9, and CXCL10 by ELISA (R&D systems). Total protein in each sample was determined by DC protein assay (Bio-rad) according to kit protocol, and IFNγ, CXCL9, and CXCL10 levels were expressed as pg protein of interest per μg of total protein. The detection limit of IFNγ, CXCL9, and CXCL10 ELISA assay was 31.3, 15.6, and 62.5 pg/ml, respectively.

**Intracellular staining of IFNγ production in leukocyte subsets**

Processed splenocytes were washed once in complete medium, resuspended at $6 \times 10^5$ cells in 100 μl of complete medium with 1× Brefeldin A (BD Biosciences), and incubated for 3 h at 37°C. Subsequently, cells were stained with live/dead determination dye (Invitrogen) for 20 min followed by surface marker staining [α-CD45 (1:400), α-CD3 (1:200), α-CD4 (1:400), α-CD8 (1:400), α-NK1.1 (1:200), αCD11b (1:400), and αLy6G (1:400)]. Labeled cells were then fixed and permeabilized in cytoFix/Perm solution (BD Biosciences) for 20 min at 4°C before staining with α-IFNγ (clone XMG1.2, ebioscience, 1:200 dilution) in cytoFix/Perm buffer for 30 min. Cells were then washed and resuspended for flow cytometry data acquisition.

**Statistical analysis**

All statistical analyses were performed according to the appropriate test depending of the parametric or nonparametric distribution of the data using Prism 6 (GraphPad Software). Test of normality was done by D'Agostino-Pearson omnibus normality test. All data with normal distribution were analyzed by unpaired t-test or one-way ANOVA with Tukey's post-test. All data that do not meet the normality requirements were analyzed with Mann–Whitney two-tailed analysis or Kruskal–Wallis with Dunn's multiple comparisons. *P*-values < 0.05 were considered statistically significant.

**Expanded View** for this article is available online.

## Acknowledgements

We thank Khairunnisa Abdul Ghaffar, Eleanor Pang, and Zhisheng Her for assistance in the animal studies. We also thank Anis Larbi and the SIgN Flow Cytometry core for assistance with cytometry analyses and the SIgN mouse core for support in animal breeding. This research was funded by SIgN, A*STAR and supported by the Biomedical Research Council, A*STAR. Wendy W.L. Lee is supported by the postgraduate scholarship from the NUS Graduate School for Integrative Science and Engineering. The funders had no role in study design, data collection and analysis, decision to publish, or preparation of the manuscript. All authors have no conflict of interest. This project is funded by Agency for Science, Technology and Research (A*STAR) core grant.

## Author contributions

T-HT, LFPN, and LR conceived and supervised the study. T-HT, SYG, CC, CMP, SWH, WWLL, and F-ML conducted the experiments and performed data analysis. T-HT, SWH, CC, LR, and LN wrote the manuscript.

## Conflict of interest

The authors declare that they have no conflict of interest.

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
