## [Review Process File · EMBO Molecular Medicine]

Co-infection with Chikungunya virus alters trafficking of pathogenic CD8⁺ T cells into the brain and prevents *Plasmodium*-induced neuropathology

Teck-Hui Teo, Shanshan W Howland, Carla Claser, Sin Yee Gun, Chek Meng Poh, Wendy WL Lee, Fok-Moon Lum, Lisa FP Ng, Laurent Rénia

Corresponding author: Laurent Rénia, Singapore Immunology Network

Review timeline:

Submission date:	10 April 2017
Editorial Decision:	11 May 2017
Revision received:	01 September 2017
Editorial Decision:	04 October 2017
Revision received:	05 October 2017
Accepted:	10 October 2017

Transaction Report:

Editor: Céline Carret

1st Editorial Decision

11 May 2017

Thank you for the submission of your manuscript to EMBO Molecular Medicine. We have now heard back from the three referees whom we asked to evaluate your manuscript.

You will see from the comments below that while the study is of interest, referees have two main overlapping issues with it: 1) mechanism and 2) clinical relevance. While the 1st part can be evaluated experimentally especially providing the detailed suggestions of referee 1, the latter is a lot trickier. After discussing internally about the options, including with our chief editor, should you be able to address these criticisms in full, we would be willing to consider a revised manuscript, providing that the mechanism of protection organized from the spleen is experimentally deciphered and the clinical relevance better explained.

Revised manuscripts should be submitted within three months of a request for revision; they will otherwise be treated as new submissions, except under exceptional circumstances in which a short extension is obtained from the editor. Please note that EMBO Molecular Medicine encourages a single round of revision and that, as acceptance or rejection of the manuscript will depend on another round of review, your responses should be as complete as possible.

I look forward to receiving your revised manuscript.

***** Reviewer's comments *****

Referee #1 (Comments on Novelty/Model System):

model is fine.

Referee #1 (Remarks):

This manuscript by Teo and colleagues examines the influence of the arthropod-borne alpha virus Chikungunya on malaria disease. The overlap of Chikungunya and Plasmodium vectors in Asia and Africa as well as evidence from patient cohorts of Chikungunya and malaria co-infection form the rationale for this study. Utilizing the Plasmodium berghei ANKA (PbA) experimental cerebral malaria (ECM) model, the authors examine the effect of Chikungunya on the onset and severity of ECM and make some interesting observation. Concurrent PbA and CHIKV infection protects mice from neurological symptoms of ECM and ECM associated death. Moreover, the PbA + CHIKV co-infected mice sequester fewer parasites in the brain. However, the most interesting piece of data is the observation that PbA + CHIKV co-infected mice retain parasite specific CD8 T cells in the spleen and these CD8 T cells are impaired in their ability to traffic to the brain thus protecting the mice from ECM associated pathology. The retention of CD8 T cells correlated with reduced CXCR3 expression and increased IFN γ , CXCL9 and CXCL10 expression in the spleens of PbA + CHIKV co-infected mice. As such the authors propose a mechanism by which IFN γ production by CD4 T cells leads to the production of CXCL9 and CXCL10 in the spleen which in turn promotes the retention of the CD8 T cells protecting mice from ECM. This hypothesis is intriguing but the data presented in the study fall short of conclusively supporting it. For this manuscript to be suitable for publication the authors should address the comments and concerns outlined below

Major Concerns:

- 1) Does either CD4 T cell depletion or IFN γ deficiency recapitulate the ECM pathology observed in singly-PbA infected mice?
- 2) The data in Figure 5A is intriguing. One would expect that CD8 T cells from PbA + CHIKV infected mice would be able to traffic to the brain when adoptively transferred into singly-infected PbA mice as the high cytokine and chemokine index in the spleen of PbA + CHIKV mice does not exist in the PbA mice. The authors reason that the high IFN γ /CXCL9/CXCL10 index in the PbA + CHIKV mouse drives a reduction in CXCR3 expression such that even when these CD8 T cells are transferred into PbA infected mice, they still retain a trafficking defect. To bolster this argument, the authors should infect CXCL10/CXCL9 (CXCL10 KO mice are commercially available and antibodies to block CXCL9 also exist) deficient mice, and adoptively transfer these CD8 T cells into a singly-infected PbA infected mice. These CD8 T cells should not have a reduction in CXCR3 expression and should home to the livers of PbA infected mice at the same frequency as CD8 T cells from PbA infected donor mice.
- 3) It is intriguing that only acute CHIKV infection that is concurrent with a PbA infection relieves the pathology of ECM. The authors need to better examine the kinetics of IFN γ expression and viral titers in the other models (i.e Fig 1B- Fig 1D). Does the concurrent PbA + CHIKV infection afford protection from ECM simply because the peak of IFN γ induction in the spleen coincides with the expected onset of ECM? How does this compare to CHIKV infection in humans and cerebral malaria in humans especially since both acute and chronic CHIKV infection have been shown to induce very high levels of IFN γ .
- 4) The Figure legend titles for Figure 6 imply a causal relationship between CXCL9/CXCL10 and CXCR3 expression levels. However this statement is not supported by the data. Until the experiments outlined in major concern #2 are carried out, the authors need to amend this statement to show that the evidence supports a correlative relationship between CXCL9/CXCL10 levels in the

spleen and CXCR3 expression on splenic CD8 T cells in PbA + CHIKV co-infected mice.

Minor Concerns:

- 1) In figure 3 panel A, presentation is misspelt as 'presenation'
- 2) In the description of Figure 3A, line 154 to line 157, the authors state " Using a reporter cell line to measure the level of in vivowe showed that co-infection abolished cross-presentation (Fig 3A). This statement is too strong given that the data indicates that PbA +CHIKV results in a reduction in cross-presentation and not an abolishment as the authors state unless there is no statistical difference between naïve and PbA + CHIKV mice.
- 3) In Fig 3B, the panel should be labeled or the ticks at the bottom of the graph should be removed. It is confusing to the reader.
- 4) Figure 6A and 6B should have their own legends.
- 5) The authors use 'Leukocyte' and 'Leucocyte' interchangeably throughout this manuscript. While this reviewer recognizes that these are alternate spelling the authors should just stick to one spelling in the interest of uniformity
- 6) Are the data in Figure 1B-Figure 1D based on one experiment or have they been repeated?

Referee #2 (Remarks):

This study determines the impact of Chikungunya co-infection on a model of cerebral malaria (*P. berghei* ANKA infections of C57BL/6 mice). The authors have determined that simultaneous Chikungunya co-infection prevents death of mice from experimental cerebral malaria, a phenomenon that appears to derive principally from impaired cross-presentation of Plasmodium peptides by brain endothelial cells, impaired T cell trafficking to the brain and consequently a more intact blood brain barrier in co-infected animals. The authors show that co-infection is also associated with increased splenomegaly and T cells reactive to Plasmodium peptide expand but are retained in the spleen, an observation that is correlated with increased splenic expression of the chemokines CXCL9 and CXCL10.

This study is generally well presented and interesting. However a general relevance of such a study is lacking - the authors make do cite published studies on Plasmodium-Chikungunya co-infected patients but this is a relatively rare phenomenon. Furthermore it is not clear how this knowledge would substantially alter treatment regimens.

The authors present a diagram that is supposed to summarize the mechanism by which their results can be explained. Whilst this is helpful, I think they should perhaps revise the parasite load in the brain (I question whether the significant difference in parasite load in Figure 2 is biologically significant given that parasitemia is reversed 1 day later). The contribution of IFN- γ production by NK, NKT and neutrophils is questionable from the results presented in Figure 7.

The figure legends divulge the mean differences on the statistics. Whilst this is commendable with respect to data disclosure, I don't think it is necessary. However the authors maybe should double-check their statistics in some places. In Figure 5 they have used non-parametric Mann Whitney test. However I am surprised their some of the significant data reached statistical significance given that MWU calculates based on ranks. For example in the comparison of Pb1-specific CD8⁺ T cells it looks to me like there is a spread with the elevated cell number largely driven by one animal. I think some additional replicates to assess whether this is a replicable observation should be done.

Graphs should be consistent - Figure 3 has a mixture of bar graphs and dots. I liked the use of dots showing individual animals and it was not clear why the authors selected bar graphs for some of the data in this figure.

The data in Figure 5 on the in vivo migration assay may benefit from an explanatory diagram to show the experimental set up. The flow plots in Supplementary Figure 1C showing lower expression of CXCR3 should be included in figure 5

Figure 6 legend mentions "mg" of protein - I think this is a typo as the legend states it is pg/ μ g of tissue

In Figure 7 the use of the same y-axis to represent scale I think is not displaying the main message

that IFN-g comes predominantly from T cells. The lack of IFN-g from NK cells at day 6 PI is really a little puzzling. The methods section that suggests that the intracellular assays in this figure were performed with brefeldin only (no in vitro stimulation). Flow plots should be included in the figure (or in supplementary) showing the IFN-g staining in these populations and demonstration the identification of the different populations. It is also not discussed how simultaneous co-infection could drive enhanced IFN-g in the spleen.

Referee #3 (Remarks):

The report by Renia and colleagues examines the effect of co-infection with chikungunya virus on the induction of experimental cerebral malaria (ECM) in *Plasmodium berghei* ANKA infected mice. It shows that simultaneous infection is able to prevent induction of ECM and the authors provide evidence that this is largely a consequence of increase IFN γ production in the spleen, which causes induction of CXCR3-binding chemokines CXCL9 and CXCL10. Excessive levels of these chemokines in the spleen then limit somewhat the egress of parasite-specific T cells from the spleen to the brain, impairing their effects in this tissue. Overall, the data are largely supportive of the conclusions and these conclusions appear to be novel. The extent to which these findings are of interest to a broad audience may be an issue, however.

It was not proven that IFN γ was responsible for the induction of CXCR3-binding chemokines. Could this be tested in an IFN γ ko mouse?

I have a few minor concerns to be addressed.

1. Line 111. "after" should be replaced with "before".
2. Line 156. "Abolished" is a bit strong as there is some response. Perhaps "greatly reduced" would be more appropriate.
3. Line 163. It would be useful to show LFA-1 staining somewhere as my own impression is that all CD8 T cells express LFA-1, at either low or high levels. Could this be clarified for both CD4 and CD8 T cells or a reference to the details provided?
4. Line 235. Replace "was" with "were"
5. Please clarify whether IFN γ is the sole inducer of CXCL9 and 10. I am concerned that IFN α might induce these chemokines directly and be responsible for the effects seen.
6. Please clearly indicate how many time experiments were performed. This is only indicated in a few parts of a few figures. I have assumed they were done more than once, but this is not clear.

1st Revision - authors' response

01 September 2017

Referee #1

(Comments on Novelty/Model System):

model is fine.

Response: We thank the reviewer for the positive comments.

(General comments for Author):

This manuscript by Teo and colleagues examines the influence of the arthropod-borne alpha virus Chikungunya on malaria disease. The overlap of Chikungunya and *Plasmodium* vectors in Asia and Africa as well as evidence from patient cohorts of Chikungunya and malaria co-infection form the rationale for this study. Utilizing the *Plasmodium berghei* ANKA (PbA) experimental cerebral malaria (ECM) model, the authors examine the effect of Chikungunya on the onset and severity of ECM and make some interesting observation. Concurrent PbA and CHIKV infection protects mice from neurological symptoms of ECM and ECM associated death. Moreover, the PbA + CHIKV co-infected mice sequester fewer parasites in the brain. However, the most interesting piece of data is the observation that PbA + CHIKV co-infected mice retain parasite specific CD8 T cells in the

spleen and these CD8 T cells are impaired in their ability to traffic to the brain thus protecting the mice from ECM associated pathology.

Response: We thank the reviewer for the positive comments.

The retention of CD8 T cells correlated with reduced CXCR3 expression and increased IFN γ , CXCL9 and CXCL10 expression in the spleens of PbA + CHIKV co-infected mice. As such the authors propose a mechanism by which IFN γ production by CD4 T cells leads to the production of CXCL9 and CXCL10 in the spleen, which in turn promotes the retention of the CD8 T cells protecting mice from ECM. This hypothesis is intriguing but the data presented in the study fall short of conclusively supporting it. For this manuscript to be suitable for publication the authors should address the comments and concerns outlined below

Response: We have now included new concurrent co-infection experiments in IFN γ -deficient mice to better demonstrate the causal relationship between IFN γ , CXCL9/CXCL10 induction and CD8+ T cells retention in the spleen. Specific datasets will be shown to respond to the specific pointers below.

Major Concerns:

1) Does either CD4 T cell depletion or IFN γ deficiency recapitulates the ECM pathology observed in singly-PbA infected mice?

Response: We performed co-infection in IFN γ -deficient mice as recommended and as expected, ECM could not be recapitulated because PbA-infected IFN γ -deficient mice are protected from ECM. The protection of ECM in PbA-infected IFN γ deficient mice was similar to a previous report where IFN γ were shown to be a core mediator of ECM (Claser et al, 2011). Specifically, late production of IFN γ at day 5 is essential for brain endothelial cells and cross-presentation to pathogenic malaria-specific CD8+ T cells in the brain (Howland et al, 2013).

Nonetheless, we have included this new data in the revised manuscript as the new Figure S6.

Supplementary Figure S6. IFN γ ^{-/-} mice are protected ECM during single PbA or concurrent co-infection. (A) Parasitemia and (B) mortality of WT+PbA ($n = 5$), WT+PbA+CHIKV ($n = 5$), IFN γ ^{-/-}+PbA ($n = 5$) and IFN γ ^{-/-}+PbA+CHIKV ($n = 5$). Data of WT+PbA+CHIKV and IFN γ ^{-/-}+PbA+CHIKV were compared by Mann-Whitney 2-tailed analysis (** $P < 0.01$).

The amendments to the text can be found in line 293 – line 297 (Page 14):

“To further demonstrate the causal relationship between IFN γ and CXCL9/CXCL10 induction in the co-infected mice, we performed co-infection in IFN γ ^{-/-} mice. Similar to previous study, ECM does not occur in IFN γ ^{-/-} mice (Claser et al, 2011), hence ECM was not recapitulated in the co-infected IFN γ ^{-/-} mice (new Supplementary Fig S6).”

2) The data in Figure 5A is intriguing. One would expect that CD8 T cells from PbA + CHIKV infected mice would be able to traffic to the brain when adoptively transferred into singly-infected PbA mice as the high cytokine and chemokine index in the spleen of PbA + CHIKV mice does not exist in the PbA mice. The authors reason that the high IFN γ /CXCL9/CXCL10 index in the PbA + CHIKV mouse drives a reduction in CXCR3 expression such that even when these CD8 T cells are transferred into PbA infected mice, they still retain a trafficking defect. To bolster this argument, the authors should infect CXCL10/CXCL9 (CXCL10 KO mice are commercially available and antibodies to block CXCL9 also exist) deficient mice, and adoptively transfer these CD8 T cells into a singly-infected PbA infected mice. These CD8 T cells should not have a reduction in CXCR3 expression and should home to the livers of PbA infected mice at the same frequency as CD8 T cells from PbA infected donor mice.

Response: The reviewer has raised an interesting point and we have taken an alternative approach to block the effects of CXCL9/CXCL10 action during co-infection. Instead of antibody neutralization (neutralizing antibodies are not commercially available) or the use of CXCL9/CXCL10-deficient mice, co-infection in IFN γ -deficient mice was performed. Our prior data suggest that the core inducer of CXCL9/CXCL10 at the time-point of interest (6 dpi) would be IFN γ and not type I IFN (Carter et al, 2007; Groom & Luster, 2011; Teo et al, 2015). Using spleen lysates from co-infected IFN γ -deficient mice, our data also showed a significant suppression (~80 to 90%) of CXCL9 and CXCL10 in the spleen. Importantly, the loss of CXCL9/CXCL10 in co-infected IFN γ -deficient mice recapitulated CXCR3 expression and migration capacity of isolated CD8+ T cells to the brain.

These new data are now presented in the new Figure 7.

Figure 7. Elevated splenic IFN γ during co-infection induce splenic CXCL9/CXCL10 and suppress CXCR3 expression on CD8+ T cells to limit migration capacity towards the brain.

A Levels of IFN γ protein in the spleen of naïve, PbA and PbA+CHIKV groups on 2, 4 and 6 dpi ($n \geq 5$ per group). Each data point was obtained from 1 mouse. Data comparison between PbA and PbA+CHIKV groups were done by Mann-Whitney 2-tailed analysis (6 dpi; $*P = 0.0317$).

B, C Levels of CXCL9 and CXCL10 protein in the spleen of WT naïve ($n = 5$), WT+PbA ($n = 5$), WT+PbA+CHIKV ($n = 5$), IFN $\gamma^{-/-}$ +PbA ($n = 4$) and IFN $\gamma^{-/-}$ +PbA+CHIKV ($n = 4$) on 6 dpi. Data comparison between PbA and PbA+CHIKV groups in the respective WT and IFN $\gamma^{-/-}$ background were done by Mann-Whitney 2-tailed analysis (CXCL9 – WT+PbA versus WT+PbA+CHIKV; $*P = 0.0238$, IFN $\gamma^{-/-}$ +PbA versus IFN $\gamma^{-/-}$ +PbA+CHIKV; $^{ns}P = 0.7429$. CXCL10 – WT+PbA versus WT+PbA+CHIKV; $**P = 0.0079$, IFN $\gamma^{-/-}$ +PbA versus IFN $\gamma^{-/-}$ +PbA+CHIKV; $^{ns}P = 0.3429$).

D Number of total CD8+ T cells and Pb1-specific CD8+ T cells in the spleen of WT+PbA ($n = 5$), WT+PbA+CHIKV ($n = 4$), IFN $\gamma^{-/-}$ +PbA ($n = 5$) and IFN $\gamma^{-/-}$ +PbA+CHIKV ($n = 6$) on 6 dpi. Data comparison between PbA and PbA+CHIKV groups in the respective WT and IFN $\gamma^{-/-}$ background were done by Mann-Whitney 2-tailed analysis (Total CD8+ T cells - WT+PbA versus WT+PbA+CHIKV; $*P = 0.0159$, IFN $\gamma^{-/-}$ +PbA versus IFN $\gamma^{-/-}$ +PbA+CHIKV; $^{ns}P = 0.0823$, Pb1-specific CD8+ T cells - WT+PbA versus WT+PbA+CHIKV; $*P = 0.0317$, IFN $\gamma^{-/-}$ +PbA versus IFN $\gamma^{-/-}$ +PbA+CHIKV; $^{ns}P = 0.4286$).

E, F CXCR3 surface expression on total CD8+ T cells and Pb1-specific CD8+ T cells in the spleen of WT+PbA ($n = 5$), WT+PbA+CHIKV ($n = 4$), IFN $\gamma^{-/-}$ +PbA ($n = 5$) and IFN $\gamma^{-/-}$ +PbA+CHIKV ($n = 6$) on 6 dpi. Representative histograms showing CXCR3 expression was shown. Threshold of CXCR3+ cells is delineated by black dotted line. Data comparison between PbA and PbA+CHIKV groups in the respective WT and IFN $\gamma^{-/-}$ background were done by Mann-Whitney 2-tailed analysis (Total CD8+ T cells - WT+PbA versus WT+PbA+CHIKV; $*P = 0.0159$, IFN $\gamma^{-/-}$ +PbA versus IFN $\gamma^{-/-}$ +PbA+CHIKV;

^{ns}P = 0.0823, Pb1-specific CD8+ T cells - WT+PbA versus WT+PbA+CHIKV; *P = 0.0159, IFN γ ^{-/-}+PbA versus IFN γ ^{-/-}+PbA+CHIKV; ^{ns}P = 0.9307).

G *In vivo* migration assay measuring the migratory capacity of LFA-1+ and Pb1-specific CD8+ T cells from IFN γ ^{-/-}+PbA donors ($n = 4$) and IFN γ ^{-/-}+PbA+CHIKV donors ($n = 3$) towards the brain of WT PbA recipients. 7×10^6 isolated donors' CD8+ T cells (6 dpi) were transferred into PbA recipient at 5 dpi and harvested 22 hrs post transfer. All data are expressed as ratio of recovered cells to initial numbers of cell transferred into the recipients for each specific cell type. Mann-Whitney 2-tailed analysis (LFA-1+CD8+ T cells: ^{ns}P = 0.9999, Pb1-specific CD8+ T cells: ^{ns}P = 0.6286). Data information: For all cytokines or chemokines, quantifications were measured by ELISA using cell lysate from the organ and determined as pg per μ g of total protein. Each data point shown in the dot plots was obtained from 1 mouse.

The text has also been edited substantially from lines 297 – 306 (Page 14) to read:

“Induction of splenic CXCL9 (~10 fold reduction) and CXCL10 (~4 fold reduction) was abolished in co-infected IFN γ ^{-/-} mice (Fig 7B and C), and restored the numbers of splenic total and Pb1-specific CD8+ T cells. CXCR3 expression in the co-infected IFN γ ^{-/-} mice was similar to single PbA-infected mice (Fig 7D–F). Importantly, when isolated CD8+ T cells (6 dpi) from single PbA-infected or co-infected donors in IFN γ ^{-/-} background were transferred into WT PbA-infected recipients, a similar proportion of LFA-1+ and Pb1-specific CD8+ T cells was recovered in the recipient brains (Fig 7G), suggesting a restoration of migration capacity of these CD8+ T cells in the IFN γ ^{-/-} co-infected donor.”

3) It is intriguing that only acute CHIKV infection that is concurrent with a PbA infection relieves the pathology of ECM. The authors need to better examine the kinetics of IFN γ expression and viral titers in the other models (i.e Fig 1B- Fig 1D). Does the concurrent PbA + CHIKV infection afford protection from ECM simply because the peak of IFN γ induction in the spleen coincides with the expected onset of ECM? How does this compare to CHIKV infection in humans and cerebral malaria in humans especially since both acute and chronic CHIKV infection have been shown to induce very high levels of IFN γ .

Response: We have taken the reviewer's suggestion to profile the induction of IFN γ in the spleen during CHIKV infection. As expected, IFN γ was only induced on 6 dpi and not earlier. This timing also corroborated with the induction of IFN γ -producing CHIKV-specific CD4+ T cells in the spleen during single-CHIKV infection (Teo et al, 2017). We wish to highlight that IFN γ is an acute pro-inflammatory cytokine and their levels are not elevated during the chronic phase of the disease (Teng et al, 2015; Teo et al, 2015).

The IFN γ induction profile in the spleen during CHIKV is now presented in the new Fig S5.

Supplementary Figure S5. IFN γ induction profile in the spleen during CHIKV infection. IFN γ level in the cell lysate from the spleen of naïve and CHIKV-infected mice on 2 dpi, 4 dpi and 6 dpi ($n \geq 4$). IFN γ level in the cell lysate were determined using IFN γ ELISA and expression level was normalized relative to the respective naïve mice harvested in the same experiment. Each data point in the dot plot represent data collected from 1 mouse.

The amendments can be found in the result section, lines 287 – 288 (Page 13) in the revised manuscript to read “In addition, we showed that IFN γ was induced in the spleen of single-CHIKV infected mice at 6 dpi (Supplementary Fig S5).”

And the discussion section, line 352 – 354 (Page 17) to read “The higher IFN γ are likely generated by IFN γ -producing CHIKV-specific CD4⁺ T cells in the spleen which was previously shown to be elevated on 6 days post CHIKV infection (Teo et al, 2017).”

4) The Figure legend titles for Figure 6 imply a causal relationship between CXCL9/CXCL10 and CXCR3 expression levels. However this statement is not supported by the data. Until the experiments outlined in major concern #2 are carried out, the authors need to amend this statement to show that the evidence supports a correlative relationship between CXCL9/CXCL10 levels in the spleen and CXCR3 expression on splenic CD8 T cells in PbA + CHIKV co-infected mice.

Response: We note the reviewer’s concerns. We believe that we have now responded to this in pointer 2. The absence of IFN γ during co-infection abrogated CXCL9/CXCL10 in the spleen and restored CXCR3 expression level on total and parasite-specific (Pb1-specific) CD8⁺ T cells.

Minor Concerns:

1) In figure 3 panel A, presentation is misspelt as 'presention'

Response: We apologised for this oversight and have amended it accordingly.

2) In the description of Figure 3A, line 154 to line 157, the authors state " Using a reporter cell line to measure the level of in vivowe showed that co-infection abolished cross-presentation (Fig 3A). This statement is too strong given that the data indicates that PbA +CHIKV results in a reduction in cross-presentation and not an abolishment as the authors state unless there is no statistical difference between naïve and PbA + CHIKV mice.

Response: The word “abolished” has now been changed to “greatly reduced”. The sentence now reads: “we showed that co-infection greatly reduced cross-presentation (Fig 3A).”

3) In Fig 3B, the panel should be labeled or the ticks at the bottom of the graph should be removed. It is confusing to the reader.

Response: Fig 3B has now been amended. The bar graph shown previously in Fig 3B has also been changed to dot-plots for consistency. For easy reference, Fig 3B is shown below.

Figure 3B: Number of CD45⁺ cells in the brain of naïve ($n = 8$), CHIKV ($n = 7$), PbA ($n = 6$) and PbA+CHIKV ($n = 5$) groups on 6 dpi. Data shown were representative of two independent experiments. All data were analyzed by one-way ANOVA with Tukey’s post-test. Naïve versus PbA; ***mean diff* = -142082, CHIKV versus PbA; ***mean diff* = -145113, PbA versus PbA+CHIKV; **mean diff* = 120725.

4) Figure 6A and 6B should have their own legends.

Response: Figure legend has been changed accordingly.

A Levels of CXCL9 protein in the spleen of naïve, PbA and PbA+CHIKV groups on 2, 4 and 6 dpi ($n \geq 5$ per group). Each data point was obtained from 1 mouse. Data comparison between PbA and PbA+CHIKV groups were done by Mann-Whitney 2-tailed analysis (6 dpi; $**P = 0.0079$). Data shown were pooled from 3 independent experiments. Chemokine protein levels were determined as pg per μg of total protein, measured by ELISA using cell lysate from the spleen.

B Levels of CXCL10 protein in the spleen of naïve, PbA and PbA+CHIKV groups on 2, 4 and 6 dpi ($n \geq 5$ per group). Each data point was obtained from 1 mouse. Data comparison between PbA and PbA+CHIKV groups were done by Mann-Whitney 2-tailed analysis (6 dpi; $*P = 0.0317$). Data shown were pooled from 3 independent experiments. Chemokine protein levels were determined as pg per μg of total protein, measured by ELISA using cell lysate from the spleen.

5) The authors use 'Leukocyte' and 'Leucocyte' interchangeably throughout this manuscript. While this reviewer recognizes that these are alternate spelling the authors should just stick to one spelling in the interest of uniformity

Response: Manuscript has been checked for consistency, and all “leukocytes” have now been edited to “leucocytes”.

6) Are the data in Figure 1B-Figure 1D based on one experiment or have they been repeated?

Response: Data shown were representative of 2 independent experiments. The figure legends have been updated to reflect this information.

Referee #2 (Remarks):

1. This study determines the impact of Chikungunya co-infection on a model of cerebral malaria (*P. berghei* ANKA infections of C57BL/6 mice). The authors have determined that simultaneous Chikungunya co-infection prevents death of mice from experimental cerebral malaria, a phenomenon that appears to derive principally from impaired cross-presentation of Plasmodium peptides by brain endothelial cells, impaired T cell trafficking to the brain and consequently a more intact blood brain barrier in co-infected animals. The authors show that co-infection is also associated with increased splenomegaly and T cells reactive to Plasmodium peptide expand but are retained in the spleen, an observation that is correlated with increased splenic expression of the chemokines CXCL9 and CXCL10.

Response: We thank the reviewer for the positive comments.

2. This study is generally well presented and interesting. However, a general relevance of such a study is lacking - the authors make do cite published studies on Plasmodium-Chikungunya co-infected patients but this is a relatively rare phenomenon.

Response: The reviewer raised an important statement. We note that despite low number of epidemiology reports (Ayorinde et al, 2016; Baba et al, 2013; Chipwaza et al, 2014; Hertz et al, 2012; Waggoner et al, 2017), the phenomenon of such co-infections may not be as rare as the reviewer suggested. Specifically, it is still not a standard practice to screen for evidence of exposure to chikungunya virus by serology or molecular methods in malaria patients. The lack of proper screening could explain the lack of co-infection reports. To illustrate this, we have previously screened for CHIKV-specific antibodies in the plasma of a small cohort (n = 52) of acute *P. vivax* patients collected in late 2011 from Mae Sot, Thailand. Interestingly, 36.54% of these patients showed positive detection of CHIKV-specific IgM and 88.46% showed detectable IgG, demonstrating the presence of both co-acute and sequential co-infection. These data were not included in the manuscript as they were outside the scope of the paper. Nonetheless, we present the unpublished data below for review purposes.

Legend: CHIKV-specific IgM and Total IgG titer in acute *P. vivax* patients. CHIKV-specific antibodies were determined using virion-based ELISA (Kam et al, 2012) with patients' plasma (n = 52). CHIKV-specific IgM were quantified at 1:100 dilution. CHIKV-specific IgG were quantified at 1:2000 dilution. Antibody positive patients were determined by OD reading above mean healthy controls + 6 SD (n=3). Percentage representation in pie charts was shown beside dot plot.

Furthermore, it is not clear how this knowledge would substantially alter treatment regimens

The increase incidences of chikungunya and other arboviruses would mean an increase of co-infection with other pathogens. While the protective effect of co-infection between *Plasmodium* and chikungunya virus do not alter treatment regime of malaria, similar immune regulation by chikungunya virus with other pro-inflammatory infectious disease could be detrimental and not protective. This could help clinicians focus on the right mediators when understanding complications of chikungunya co-infection with other diseases in the population.

The discussion in lines 386 - 405, page 18 has been written to emphasize on the clinical relevance of this study:

“While co-infection with CHIKV is protective in the context of ECM, co-infection of CHIKV with other pathogens could be detrimental if CD8⁺ T cells are needed for protection in the peripheral tissues. In contrast to their pathogenicity in ECM, CD8⁺ T cells are needed for protection against a wide spectrum of neurological infections including West Nile virus (Brien et al, 2007; Klein et al, 2005), human simplex virus-1 (Lang & Nikolich-Zugich, 2005), cytomegalovirus (Bantug et al, 2008), enterovirus 71 (Lin et al, 2009), *Toxoplasma gondii* (Guiton et al, 2009), neuroborreliosis (Jacobsen et al, 2003) and central nervous system listeriosis (Hayashi et al, 2009; Schluter et al, 1995). The IFN γ -dependent mechanism described here induced by co-infection with CHIKV could possibility alter or delay CD8⁺ T cell migration to the brain, leading to disease aggravation. The ECM results here underscore the importance of further studies into the regulation and function of T cell migration in the context of CHIKV co-infection with other pathogens. Such co-infections may interfere with protective or pathogenic immune responses induced by single infection.

Understanding the interfering mechanisms will pave the way in the development of new therapeutic modalities. Lastly, it also calls for comprehensive pathogen identification in endemic countries when symptomatic diagnosis of one pathogen may be confounded by another pathogen.”

3. The authors present a diagram that is supposed to summarize the mechanism by which their results can be explained. Whilst this is helpful, I think they should perhaps revise the parasite load in the brain (I question whether the significant difference in parasite load in Figure 2 is biologically significant given that parasitemia is reversed 1 day later).

Response: We wish to highlight that parasite load quantified in Fig 2 reflected sequestered parasites in the brain, while parasitemia is the measurement of parasites in the circulation (Fig 1). It has been shown that the presence of sequestered parasites in the brain at the time of ECM induction (6 dpi) is essential for induction of pathogenic brain endothelia cell cross-presentation (Howland et al, 2015). Therefore, the unique drop in sequestered parasites in the brain at 6 dpi in the co-infected mice explains the lack of brain endothelia cells cross-presentation. This hypothesis is highlighted in the mechanism summary.

Importantly, the sequestered parasites in co-infected and single PbA infected mice were similar from 7 dpi onwards because 4 out of 7 mice died of ECM in the single PbA infected group. As such, data from these 4 mice (which are more severe, with higher sequestered parasites) were not captured from 7 dpi onwards. Instead, data captured in the single PbA-infected mice from 7 dpi onwards were non-ECM mice, which is expected to have the same level of sequestered parasites as the non-ECM mice in the co-infected group. For easy reference to the mortality curve, the data is presented below and also found in Fig 1A.

Legend: Mortality curve of PbA ($n = 7$) and PbA+CHIKV ($n = 7$) groups extracted from Fig 1A.

4. The contribution of IFN- γ production by NK, NKT and neutrophils is questionable from the results presented in Figure 7.

Response: We thank the reviewer for highlighting this and we have now re-analysed the data on IFN γ producing cells in the spleen during infection. As shown, IFN γ could be detected in CD4 $^+$ T cells, CD8 $^+$ T cells, NK cells, NKT and neutrophils (Fig 8A). However, only CD4 $^+$ T cells showed the most prominent induction in IFN γ expression level and increase in IFN γ^+ numbers upon co-infection (Fig 8). NK cells were reduced in the spleen with co-infection, while NKT cell numbers remained as a minority subset in comparison to others. Thus, NK cells are unlikely to be the major contributors of IFN γ in the spleen. For neutrophils, while IFN γ expression was high in all 4 groups, the difference between single PbA infection and co-infection was a result of a change in neutrophils numbers, and not a change in IFN γ expression level. To improve the clarity, the text has been revised from lines 308 – 323, page 14 to read

“Splenic CD4 $^+$ T cells is the major contributor of IFN γ during co-infection

To identify leucocyte subsets responsible for the higher levels of splenic IFN γ during co-infection, intracellular staining for IFN γ was performed on splenocytes harvested on 6 dpi. CD4 $^+$ T cells, CD8 $^+$ T cells, NK cells, NKT cells and neutrophils were identified to be the main IFN γ producers in the spleen (Fig 8A). Interestingly, only CD4 $^+$ T cells showed the most prominent induction of IFN γ

expression and increase in IFN γ + numbers upon co-infection (Fig 8). Co-infection did not alter IFN γ expression in CD8+ T cells. Total and IFN γ -producing NK cells were reduced with parasite infection. IFN γ -producing NKT cells was a minor producing subset in comparison to other subsets (Fig 8), hence they are unlikely to be the major contributors to this phenomenon. For neutrophils, while IFN γ expression is high in all 4 groups, the difference between single PbA infection and co-infection was a result of neutrophils reduction in single PbA infection and not an increase of IFN γ expression level in the co-infected mice (Fig 8). Taken together, CD4+ T cells are the major contributors of higher splenic IFN γ during co-infection.”

For easy reference, the new Figure 8 is shown below.

Figure 8. Increased IFN γ producing CD4+ T cells in the spleen drives the enhanced splenic IFN γ on 6 dpi during concurrent co-infection.

A Representative histogram showing IFN γ production in CD4+ T cells, CD8+ T cells, NK cells, NKT cells and neutrophils in the spleen of naïve ($n = 5$), CHIKV ($n = 5$), PbA ($n = 6$) and PbA+CHIKV ($n = 8$) on 6 dpi. Black dotted line represent threshold setting for IFN γ + cells.

B Numbers of CD4+ T cells, CD8+ T cells, NK cells, NKT cells and Neutrophils in the spleen of naïve ($n = 5$), CHIKV ($n = 5$), PbA ($n = 6$) and PbA+CHIKV ($n = 8$) on 6 dpi. CD4+, CD8+ T cells, NK cells, NKT cells and neutrophils were defined as CD3+CD4+, CD3+CD8+, CD3-NK1.1+, CD3+NK1.1+ and CD3-CD11b+Ly6G+ cells respectively. All data analyzed by one-way ANOVA with Tukey's post-test. For CD4+ T cells: Naïve versus PbA+CHIKV; **mean diff* = -1.49×10^7 , PbA versus PbA+CHIKV; **mean diff* = -1.57×10^7 . For CD8+ T cells: Naïve versus PbA+CHIKV; **mean diff* = -9.61×10^6 , CHIKV versus PbA+CHIKV; **mean diff* = -9.43×10^6 , PbA versus PbA+CHIKV; **mean diff* = -9.75×10^6 . For NK cells: Naïve versus PbA; ****mean diff* = 5.22×10^6 , Naïve versus PbA+CHIKV; ***mean diff* = 4.03×10^6 , CHIKV versus PbA; ****mean diff* = 5.39×10^6 , CHIKV versus PbA+CHIKV; ***mean diff* = 4.20×10^6 . For NKT cells: Naïve versus PbA; **mean diff* = 7.95×10^5 , CHIKV versus PbA; **mean diff* = 7.34×10^5 , PbA versus PbA+CHIKV; ***mean diff* = -8.19×10^5 . For neutrophils: Naïve versus PbA; **mean diff* = 1.51×10^6 , CHIKV versus PbA; **mean diff* = 1.34×10^6 , CHIKV versus PbA+CHIKV; **mean diff* = -1.32×10^6 , PbA versus PbA+CHIKV; ****mean diff* = -2.66×10^6 .

C Numbers of IFN γ producing CD4+ T cells, CD8+ T cells, NK cells, NKT cells and Neutrophils in the spleen of naïve ($n = 5$), CHIKV ($n = 5$), PbA ($n = 6$) and PbA+CHIKV ($n = 8$) on 6 dpi. All data analyzed by one-way ANOVA with Tukey's post-test. For IFN γ +CD4+ T cells: Naïve versus PbA+CHIKV; ****mean diff* = -5.99×10^6 , CHIKV versus PbA+CHIKV; ****mean diff* = -5.52×10^6 , PbA versus PbA+CHIKV; ****mean diff* = -3.95×10^6 . For IFN γ +CD8+ T cells: Naïve versus PbA+CHIKV; **mean diff* = -9.28×10^5 , CHIKV versus PbA+CHIKV; ***mean diff* = -1.08×10^6 . For IFN γ +NK cells: Naïve versus CHIKV; ****mean diff* = 1.83×10^5 , Naïve versus PbA; ****mean diff* = 2.81×10^5 , Naïve versus PbA+CHIKV; ****mean diff* = 2.83×10^5 . For IFN γ +NKT cells: Naïve versus PbA+CHIKV; **mean diff* = -68295 , CHIKV versus PbA+CHIKV; ***mean diff* = -93362 , PbA versus PbA+CHIKV; ***mean diff* = -80084 . For IFN γ +neutrophils: Naïve versus PbA; **mean diff* = 1.58×10^6 , CHIKV versus PbA; **mean diff* = 1.40×10^6 , PbA versus PbA+CHIKV; ****mean diff* = -2.50×10^6 .

In line with this conclusion, Figure 9 has now been amended to reflect CD4+ T cells as the major producer of IFN γ as illustrated below.

Figure 9. Proposed mechanism of ECM protection during concurrent co-infection of PbA and CHIKV.

In normal PbA infection (shown on the left) in susceptible C57BL/6J, parasite-specific pathogenic CD8+ T cells are generated in the spleen. These CD8+ T cells migrate to the brain and interact with brain endothelia cross-presenting parasite proteins, leading to cytotoxic events that alters the BBB, causing eventual death. Pathogenic events in the brain were highlighted in *italics brown font*. In concurrent co-infection of PbA and CHIKV (shown on the right), increased IFN γ production was induced in splenic CD4+ T cells. This drives the enhanced production of CXCL9 and CXCL10 in the spleen. Increase CXCR3 –chemokines interaction leads to internalization and retention of CXCR3 on the pathogenic CD8+ T cells in the spleen. This reduce migration of these cells into the brain and abrogate downstream pathogenic event mediated by parasite-specific CD8+ cells. Pathogenic events suppressed during co-infection were highlighted in *grey font*.

5. The figure legends divulge the mean differences on the statistics. Whilst this is commendable with respect to data disclosure, I don't think it is necessary.

Response: The detailed statistical p value and mean differences was provided as part of the requirement stated in the “authors’ guidelines” of EMBO Mol Med.

6. However the authors maybe should double-check their statistics in some places. In Figure 5 they have used non-parametric Mann Whitney test. However I am surprised their some of the significant data reached statistical significance given that MWU calculates based on ranks. For example in the comparison of Pb1-specific CD8+ T cells it looks to me like there is a spread with the elevated cell number largely driven by one animal. I think some additional replicates to assess whether this is a replicable observation should be done.

Response: We wish to state that statistical analyses were double-checked for errors. Data shown in Fig 5A was from a pool of 2 experimental replicates. This information has now been specified in the figure legend.

7. Graphs should be consistent - Figure 3 has a mixture of bar graphs and dots. I liked the use of dots showing individual animals and it was not clear why the authors selected bar graphs for some of the data in this figure.

Response: Data from Figure 3 has been changed to dot plots as illustrated below.

Figure 3 - Concurrent co-infection prevents T cells sequestration and microvessel cross-presentation in the brain.

A Brain microvessel cross-presentation for Pb1 epitopes in naïve ($n = 5$), PbA ($n = 5$) and PbA+CHIKV ($n = 5$) groups on 6 dpi. One-way ANOVA with Tukey's post-test (Naïve versus PbA; $**$ mean diff = -922.8, PbA versus PbA+CHIKV; $**$ mean diff = 888.2).

B-D CD45+, Total and LFA-1+ CD4+ T cells, and Total and LFA-1+ CD8+ T cells in the brain of naïve ($n = 8$), CHIKV ($n = 7$), PbA ($n = 6$) and PbA+CHIKV ($n = 5$) groups on 6 dpi. Data shown were representative of two independent experiments. All data analyzed by one-way ANOVA with Tukey's post-test. For CD45+ cells: Naïve versus PbA; $**$ mean diff = -142082, CHIKV versus PbA; $**$ mean diff = -145113, PbA versus PbA+CHIKV; $*$ mean diff = 120725. For total CD4 T cells: Naïve versus PbA; $**$ mean diff = -12000, CHIKV versus PbA; $**$ mean diff = -11360. For LFA-1+CD4 T cells: Naïve versus PbA; $***$ mean diff = -7828, CHIKV versus PbA; $***$ mean diff = -7243, PbA versus PbA+CHIKV; $*$ mean diff = 4521. For total CD8 T cells: Naïve versus PbA; $***$ mean diff = -65880, CHIKV versus PbA; $***$ mean diff = -64730, PbA versus PbA+CHIKV; $**$ mean diff = 46430. For LFA-1+ CD8 T cells: Naïve versus PbA; $***$ mean diff = -57400, CHIKV versus PbA; $***$ mean diff = -56280, PbA versus PbA+CHIKV; $**$ mean diff = 39320.

E Parasite epitope (Pb1)-specific CD8+ T cells in the brain of PbA ($n = 6$) and PbA+CHIKV ($n = 5$) groups on 6 dpi. Data shown were representative of two independent experiments. Mann-Whitney 2-tailed analysis ($*P = 0.0303$).

8. The data in Figure 5 on the in vivo migration assay may benefit from an explanatory diagram to show the experimental set up.

Response: This is a good suggestion and a new Supplementary Fig S8 with an explanatory schematic diagram of the *in vivo* migration assay has been added in the revised manuscript.

Supplementary Figure S8. Schematic diagram of transfer protocol for *in vivo* migration assay. On 6 dpi, CD8⁺ T cells is isolated from the spleen of PbA donors or co-infected donors. Equal numbers of CD8⁺ T cells from each group of donors were transferred into PbA-infected recipients at 5 dpi. Recovery of CFSE-tagged transferred cells in the brain were profiled using flow cytometry.

Changes to the text in materials and methods citing Supplementary Fig S8 can be found in line 529-530, page 24 as below:

“Schematic diagram of this assay is provided in Supplementary Fig S8.”

The flow plots in Supplementary Figure 1C showing lower expression of CXCR3 should be included in figure 5

Response: The flow plot has now been added into Fig 5 as shown below.

Figure 5 - Concurrent co-infection abrogates CD8+ T cells migratory capacity to the brain and surface expression of CXCR3 in the spleen.

A *In vivo* migration assay measuring the migratory capacity of total, LFA-1+ and Pb1-specific CD8+ T cells from PbA donors ($n = 5$) and PbA+CHIKV donors ($n = 10$) towards the brain of PbA recipients. 5×10^6 isolated donors' CD8+ T cells (6 dpi) were transferred into PbA recipient at 5 dpi and harvested 22 hrs post transfer. All data are expressed as ratio of recovered cells to initial numbers of cell transferred into the recipients for each specific cell type. Mann-Whitney 2-tailed analysis (LFA-1+CD8+ T cells: $*P = 0.0193$, Pb1-specific CD8+ T cells: $*P = 0.0280$). Data shown were pooled from 2 independent experiments.

B Representative histogram of CXCR3 expression in Naïve CD8+ T cells, Pb1-specific CD8+ T cells from PbA mice and Pb1-specific CD8+ T cells from co-infected mice were shown. Dotted line represent threshold for delineating CXCR3+ cells.

C Surface expression of CXCR3 in total CD8+ T cells and Pb1-specific CD8+ T cells in the spleen of PbA ($n = 6$) and PbA+CHIKV ($n = 7$) on 6 dpi. Surface expression is determined by the geometric mean of CXCR3 signal by flowcytometry. Data shown are representative of 2 independent experiments. Mann-Whitney 2-tailed analysis (Total CD8+ T cells: $**P = 0.0012$, Pb1-specific CD8+ T cells: $**P = 0.0012$).

D Number of CXCR3+CD8+ T cells and CXCR3+Pb1+ CD8+ T cells in the spleen of PbA ($n = 6$) and PbA+CHIKV ($n = 7$) on 6 dpi. Data shown is a representative of 2 independent experiments. Mann-Whitney 2-tailed analysis (Total CD8+ T cells: $*P = 0.035$).

E, F Transwell migration assay using CXCL9 and CXCL10 with isolated total CD8+ cells from PbA ($n = 6$) and PbA+CHIKV ($n = 6$) on 6 dpi. Chemotaxis index was determined as the [cells across transwell with CXCL9 or CXCL10/cells across transwell without chemokines]. Data shown are representative of two independent experiments. Mann-Whitney 2-tailed analysis (For CXCL10 total CD8+ T cells: $**P = 0.0022$).

9. Figure 6 legend mentions "mg" of protein - I think this is a typo as the legend states it is pg/ μ g of tissue

Response: Changes has been made accordingly.

10. In Figure 7 the use of the same y-axis to represent scale I think is not displaying the main message that IFN-g comes predominantly from T cells. The lack of IFN-g from NK cells at day 6 PI is really a little puzzling. The methods section that suggests that the intracellular assays in this figure were performed with brefeldin only (no in vitro stimulation). Flow plots should be included in the figure (or in supplementary) showing the IFN-g staining in these populations and demonstration the identification of the different populations.

Response: We have now incorporated the flow plots of IFN γ staining in the new Figure 8 and shown below.

Figure 8A: Representative histogram showing IFN γ production in CD4+ T cells, CD8+ T cells, NK cells, NKT cells and neutrophils in the spleen of naïve, CHIKV, PbA and PbA+CHIKV on 6 dpi.

IFN γ staining is detectable in NK cells even in the naïve mice (~5-10%). However, the drop of IFN γ + NK cells in PbA and co-infected group was due to the loss of NK cells in the spleen arising from the parasite infection (Fig 8B).

We did the IFN γ capture experiment without re-stimulation because we wanted to profile the inherent level of IFN γ during harvest.

It is also not discussed how simultaneous co-infection could drive enhanced IFN-g in the spleen.

Response: This point has been addressed by pointer 3 from Reviewer #1.

Referee #3 (Remarks):

The report by Renia and colleagues examines the effect of co-infection with chikungunya virus on the induction of experimental cerebral malaria (ECM) in *Plasmodium berghei* ANKA infected mice. It shows that simultaneous infection is able to prevent induction of ECM and the authors provide evidence that this is largely a consequence of increase IFN γ production in the spleen, which causes induction of CXCR3-binding chemokines CXCL9 and CXCL10. Excessive levels of these chemokines in the spleen then limit somewhat the egress of parasite-specific T cells from the spleen to the brain, impairing their effects in this tissue. Overall, the data are largely supportive of the conclusions and these conclusions appear to be novel. The extent to which these findings are of interest to a broad audience may be an issue, however.

Response: We thank the reviewer for the positive comments.

It was not proven that IFN γ was responsible for the induction of CXCR3-binding chemokines. Could this be tested in an IFN γ ko mouse?

Response: As suggested by reviewer, new experiments to profile CXCL9/CXCL10 in the spleen of co-infected IFN γ -deficient mice have been done. The absence of IFN γ during co-infection abrogated

induction of CXCL9 and CXCL10 by 90% in the spleen. This information can be found in the new Figure 7.

Details of the new figure and changes in main text could be found above in the reply to pointer 2, Reviewer #1.

I have a few minor concerns to be addressed.

1. Line 111. "after" should be replaced with "before".

Response: Changes made accordingly.

2. Line 156. "Abolished" is a bit strong as there is some response. Perhaps "greatly reduced" would be more appropriate.

Response: Changes made accordingly.

3. Line 163. It would be useful to show LFA-1 staining somewhere as my own impression is that all CD8 T cells express LFA-1, at either low or high levels. Could this be clarified for both CD4 and CD8 T cells or a reference to the details provided?

Response: We have now included the gating strategy for the T cells in the brain in the new Supplementary Fig S7 to address this query. Majority of the CD4+ and CD8+ T cells are LFA-1+ as illustrated below.

Supplementary Figure S7. Representative gating strategy for T cell infiltrates in the brain on 6 dpi. Total and activated CD4+ T cells in the brain were defined by CD45hi/CD3+/CD4+ and CD45hi/CD3+/CD4+/LFA-1+ cells respectively. Total and activated CD8+ T cells in the brain were defined by CD45hi/CD3+/CD8+ and CD45hi/CD3+/CD8+/LFA-1+ cells respectively.

Changes to the main text citing Supplementary Fig S7 could be found in lines 488 – 491, page 23 to read “Gating strategy of T cells infiltrate in the brain is shown in Supplementary Fig S7. Majority of the T cells infiltrate in the brain of the infected mice express LFA-1 marker (Supplementary Fig S7).”

4. Line 235. Replace "was" with "were"

Response: Changes made accordingly

5. Please clarify whether IFN γ is the sole inducer of CXCL9 and 10. I am concerned that IFN α might induce these chemokines directly and be responsible for the effects seen.

Response: Co-infection in IFN γ -deficient mice showed that IFN γ is the major inducer of CXCL9 and CXCL10 in the spleen at the time-point of interest. This has also been addressed in our response to pointer 2 of Reviewer #1.

In addition, IFN α is unlikely to be an inducer of CXCL9 and CXCL10 in this context. Prior cytokines study showed that IFN α induced during CHIKV infection returned to basal level on 6 dpi (Teo et al, 2015), the time-point where CXCL9 and CXCL10 is induced in this current study. The original data from this previous study has been pasted below for easy reference below. Do note that the strain of virus used in this study was LR2006 OPY1.

Legend: Cytokines levels in the sera during CHIKV infection on 6 dpi. All data were extracted from Teo et al., 2016. Patterns of immune mediators by two-way hierarchical clustering were shown. Each coloured cell represents the relative levels of expression of a particular cytokine in a mouse. The level of IFN α at 6 dpi is highlighted in blue box for easy reference.

6. Please clearly indicate how many time experiments were performed. This is only indicated in a few parts of a few bla figures. I have assumed they were done more than once, but this is not clear.

Response: All experiments that were done more than once and this information has now been indicated in the respective figure legends.

References

Ayorinde AF, Oyeyiga AM, Nosegbe NO, Folarin OA (2016) A survey of malaria and some arboviral infections among suspected febrile patients visiting a health centre in Simawa, Ogun State, Nigeria. *J Infect Public Health* 9: 52-59

Baba M, Logue CH, Oderinde B, Abdulmaleek H, Williams J, Lewis J, Laws TR, Hewson R, Marcello A, P DA (2013) Evidence of arbovirus co-infection in suspected febrile malaria and typhoid patients in Nigeria. *J Infect Dev Ctries* 7: 51-59

Bantug GR, Cekinovic D, Bradford R, Koontz T, Jonjic S, Britt WJ (2008) CD8 $^{+}$ T lymphocytes control murine cytomegalovirus replication in the central nervous system of newborn animals. *J Immunol* 181: 2111-2123

- Brien JD, Uhrlaub JL, Nikolich-Zugich J (2007) Protective capacity and epitope specificity of CD8(+) T cells responding to lethal West Nile virus infection. *Eur J Immunol* 37: 1855-1863
- Carter SL, Muller M, Manders PM, Campbell IL (2007) Induction of the genes for Cxcl9 and Cxcl10 is dependent on IFN-gamma but shows differential cellular expression in experimental autoimmune encephalomyelitis and by astrocytes and microglia in vitro. *Glia* 55: 1728-1739
- Chipwaza B, Mugasa JP, Selemani M, Amuri M, Mosha F, Ngatunga SD, Gwakisa PS (2014) Dengue and Chikungunya fever among viral diseases in outpatient febrile children in Kilosa district hospital, Tanzania. *PLoS Negl Trop Dis* 8: e3335
- Claser C, Malleret B, Gun SY, Wong AY, Chang ZW, Teo P, See PC, Howland SW, Ginhoux F, Renia L (2011) CD8+ T cells and IFN-gamma mediate the time-dependent accumulation of infected red blood cells in deep organs during experimental cerebral malaria. *PLoS One* 6: e18720
- Groom JR, Luster AD (2011) CXCR3 ligands: redundant, collaborative and antagonistic functions. *Immunol Cell Biol* 89: 207-215
- Guiton R, Zagani R, Dimier-Poisson I (2009) Major role for CD8 T cells in the protection against *Toxoplasma gondii* following dendritic cell vaccination. *Parasite Immunol* 31: 631-640
- Hayashi T, Nagai S, Fujii H, Baba Y, Ikeda E, Kawase T, Koyasu S (2009) Critical roles of NK and CD8+ T cells in central nervous system listeriosis. *J Immunol* 182: 6360-6368
- Hertz JT, Munishi OM, Ooi EE, Howe S, Lim WY, Chow A, Morrissey AB, Bartlett JA, Onyango JJ, Maro VP et al (2012) Chikungunya and dengue fever among hospitalized febrile patients in northern Tanzania. *Am J Trop Med Hyg* 86: 171-177
- Howland SW, Poh CM, Gun SY, Claser C, Malleret B, Shastri N, Ginhoux F, Grotenbreg GM, Renia L (2013) Brain microvessel cross-presentation is a hallmark of experimental cerebral malaria. *EMBO Mol Med* 5: 916-931
- Howland SW, Poh CM, Renia L (2015) Activated Brain Endothelial Cells Cross-Present Malaria Antigen. *PLoS Pathog* 11: e1004963
- Jacobsen M, Zhou D, Cepok S, Nessler S, Happel M, Stei S, Wilske B, Sommer N, Hemmer B (2003) Clonal accumulation of activated CD8+ T cells in the central nervous system during the early phase of neuroborreliosis. *J Infect Dis* 187: 963-973
- Kam YW, Simarmata D, Chow A, Her Z, Teng TS, Ong EK, Renia L, Leo YS, Ng LF (2012) Early appearance of neutralizing immunoglobulin G3 antibodies is associated with chikungunya virus clearance and long-term clinical protection. *J Infect Dis* 205: 1147-1154
- Klein RS, Lin E, Zhang B, Luster AD, Tollett J, Samuel MA, Engle M, Diamond MS (2005) Neuronal CXCL10 directs CD8+ T-cell recruitment and control of West Nile virus encephalitis. *J Virol* 79: 11457-11466
- Lang A, Nikolich-Zugich J (2005) Development and migration of protective CD8+ T cells into the nervous system following ocular herpes simplex virus-1 infection. *J Immunol* 174: 2919-2925
- Lin YW, Chang KC, Kao CM, Chang SP, Tung YY, Chen SH (2009) Lymphocyte and antibody responses reduce enterovirus 71 lethality in mice by decreasing tissue viral loads. *J Virol* 83: 6477-6483
- Schluter D, Oprisiu SB, Chahoud S, Weiner D, Wiestler OD, Hof H, Deckert-Schluter M (1995) Systemic immunization induces protective CD4+ and CD8+ T cell-mediated immune responses in murine *Listeria monocytogenes* meningoencephalitis. *Eur J Immunol* 25: 2384-2391

Teng TS, Kam YW, Lee B, Hapuarachchi HC, Wimal A, Ng LC, Ng LF (2015) A Systematic Meta-analysis of Immune Signatures in Patients With Acute Chikungunya Virus Infection. *J Infect Dis* 211: 1925-1935

Teo TH, Chan YH, Lee WW, Lum FM, Amrun SN, Her Z, Rajarethinam R, Merits A, Rotzschke O, Renia L et al (2017) Fingolimod treatment abrogates chikungunya virus-induced arthralgia. *Sci Transl Med* 9

Teo TH, Her Z, Tan JJ, Lum FM, Lee WW, Chan YH, Ong RY, Kam YW, Leparc-Goffart I, Gallian P et al (2015) Caribbean and La Reunion Chikungunya Virus Isolates Differ in Their Capacity To Induce Proinflammatory Th1 and NK Cell Responses and Acute Joint Pathology. *J Virol* 89: 7955-7969

Waggoner J, Brichard J, Mutuku F, Ndenga B, Heath CJ, Mohamed-Hadley A, Sahoo MK, Vulule J, Lefterova M, Banaei N et al (2017) Malaria and Chikungunya Detected Using Molecular Diagnostics Among Febrile Kenyan Children. *Open Forum Infect Dis* 4: ofx110

2nd Editorial Decision

04 October 2017

Thank you for the submission of your revised manuscript to EMBO Molecular Medicine and apologies for the delay in getting back to you. We have now received the enclosed reports from the referees that were asked to re-assess it. As you will see the reviewers are now rather supportive except for referee1 who's making a strong case in performing one experiment that you haven't. While the rest of the revision is satisfactory, I would like to strongly encourage you to perform this experiment. In addition, please address editorial concerns.

Please submit your revised manuscript as soon as possible.

***** Reviewer's comments *****

Referee #1 (Remarks for Author):

One of my Major Concerns has not been addressed adequately and requires additional experimentation:

1) Does either CD4 T cell depletion or IFN γ deficiency recapitulate the ECM pathology observed in singly-PbA infected mice?

The authors responded to this concern by showing that IFN γ is critical to the induction of ECM and as such any infections done in IFN γ KO mice (i.e single PbA or PbA + CHIK) would be devoid of ECM making such an experiment uninformative. However, that is precisely why the CD4 T cell depletion experiments are important. In the paper cited by the authors, Claser et al 2011, the authors note that ECM is clearly dependent on IFN γ production by CD8 T cells while "other cell types like CD4+ T cells, monocytes or neutrophils or cytokines such as IL-12 and TNF- α did not influence the early increase of total parasite biomass and IRBC accumulation in different organs". In this manuscript by Teo and colleagues, the authors show similar levels of IFN γ producing CD4 and CD8 T cells in PbA infected mice but IFN γ production is significantly enhanced in the splenic CD4 population upon PbA + CHIKV infection but not in the CD8 T cell population after PbA + CHIKV infection (Figure 8). This would imply that a CD4 T cell depletion should remove the beneficial impact of IFN γ (i.e retention of pathogenic CD8 T cells) while not influencing the ability to develop ECM in the first place as CD8 T cell production of IFN γ should suffice. As such unless there is compelling evidence not to do this experiment, this reviewer recommends the CD4 T cell depletion studies be done. This experiment is the critical piece to the solidifying the conclusion that CD4 T cell production of IFN γ upon CHIKV coinfection retains CD8 T cells in the spleen and prevents their trafficking to the brain to cause ECM.

Referee #2 (Remarks for Author):

The authors have answered most of my points and those stated by the other reviewers.

Referee #3 (Comments on Novelty/Model System for Author):

The model used is fine for addressing the questions asked, but it is a matter of whether the findings are broadly relevant to human disease and therefore suitable for publication in this journal. If the editors feel the relevance is sufficient then I support publication.

Referee #3 (Remarks for Author):

The authors have addresses most concerns very well, but a question of relevance to human disease is still a concern.

2nd Revision - authors' response

05 October 2017

Reviewer #1 comments

One of my Major Concerns has not been addressed adequately and requires additional experimentation:

1) Does either CD4 T cell depletion or IFN γ deficiency recapitulate the ECM pathology observed in singly-PbA infected mice?

Response: We wish to explain that in the previous comment, the reviewer asked for CD4+ T-cells depletion or IFN γ -deficient experiment. We focused on IFN γ because this is the central mediator of the effect observed that also was raised by another reviewer.

The authors responded to this concern by showing that IFN γ is critical to the induction of ECM and as such any infections done in IFN γ KO mice (i.e. single PbA or PbA + CHIK) would be devoid of ECM making such an experiment uninformative.

Response: While the direct observation of ECM pathology is not informative in co-infected IFN γ -deficient mice, approaches have been taken to prove that IFN γ in the spleen is the core mediator for CXCR3 regulation that blocks migration. This clearly explains the protection of ECM during co-infection.

However, that is precisely why the CD4 T cell depletion experiments are important. In the paper cited by the authors, Claser et al 2011, the authors note that ECM is clearly dependent on IFN γ production by CD8 T cells while "other cell types like CD4+ T cells, monocytes or neutrophils or cytokines such as IL-12 and TNF- α did not influence the early increase of total parasite biomass and IRBC accumulation in different organs".

Response: The paper (Claser et al 2011; PLoS ONE) cited is one of our earlier studies and we believe that there is a misinterpretation of the conclusions in that report by the reviewer. In Claser et al 2011, absence of either IFN γ (IFN γ KO mice) or CD8+ T cells (by Ab-mediated depletion or CD8+ T cells KO mice) prevented the increased in total parasite biomass. However, the increase in parasite biomass was not necessarily due to IFN γ -producing CD8 T cells only.

Importantly, parasite biomass is only one of the core mediator for ECM induction, and ECM protection could occur in a mechanism independent of parasite biomass alteration. Particularly, early literature have shown that early depletion of CD4+ T cells in wild type mice prevented ECM, independent of parasite biomass reduction (Yanez et al 1996, J Immunol; Belnoue et al 2002, J Immunol; Belnoue et al 2008, parsite Immunol; Claser et al 2011, PIOS ONE). This could be due to the fact that CD4+ T cells are necessary for the full effector function of CD8+ T cells (Belnoue et al, 2002, J Immunol). Therefore to continue the explanation in point 1 above, depleting away the CD4+ T cells will prevent ECM in both the single-infected and in co-infected mice, thus rendering the experiment uninformative.

In this manuscript by Teo and colleagues, the authors show similar levels of IFN γ producing CD4 and CD8 T cells in PbA infected mice but IFN γ production is significantly enhanced in the splenic CD4 population upon PbA + CHIKV infection but not in the CD8 T cell population after PbA + CHIKV infection (Figure 8). This would imply that a CD4 T cell depletion should remove the beneficial impact of IFN γ (i.e. retention of pathogenic CD8 T cells) while not influencing the ability to develop ECM in the first place as CD8 T cell production of IFN γ should suffice.

Response: The reviewer raised an interesting point, but the assumption made is likely not possible in the context of ECM. Previous studies have demonstrated that IFN γ production from CD8+ T cells alone is not sufficient to drive ECM development (Villegas-Mendez et al, 2012). Instead, CD4+ T cells is an important source of IFN γ when driving ECM development (Villegas-Mendez et al, 2012). As such, depletion of CD4+ T cells would also remove the pathogenic effects of IFN γ required in the brain for ECM development.

As such unless there is compelling evidence not to do this experiment, this reviewer recommends the CD4 T cell depletion studies be done. This experiment is the critical piece to the solidifying the conclusion that CD4 T cell production of IFN γ upon CHIKV coinfection retains CD8 T cells in the spleen and prevents their trafficking to the brain to cause ECM.

Response: We believe we have explained the rationale in detail that performing the co-infection experiment in CD4+ T-cells-depleted condition would not bring more information to this manuscript.

References

- Belnoue E, Kayibanda M, Vigarito AM, Deschemin JC, van Rooijen N, Viguier M, Snounou G, Renia L (2002) On the pathogenic role of brain-sequestered alphabeta CD8+ T cells in experimental cerebral malaria. *J Immunol* 169: 6369-6375
- Belnoue E, Potter SM, Rosa DS, Mauduit M, Gruner AC, Kayibanda M, Mitchell AJ, Hunt NH, Renia L (2008) Control of pathogenic CD8+ T cell migration to the brain by IFN-gamma during experimental cerebral malaria. *Parasite Immunol* 30: 544-553
- Claser C, Malleret B, Gun SY, Wong AY, Chang ZW, Teo P, See PC, Howland SW, Ginhoux F, Renia L (2011) CD8+ T cells and IFN-gamma mediate the time-dependent accumulation of infected red blood cells in deep organs during experimental cerebral malaria. *PLoS One* 6: e18720
- Villegas-Mendez A, Greig R, Shaw TN, de Souza JB, Gwyer Findlay E, Stumhofer JS, Hafalla JC, Blount DG, Hunter CA, Riley EM et al (2012) IFN-gamma-producing CD4+ T cells promote experimental cerebral malaria by modulating CD8+ T cell accumulation within the brain. *J Immunol* 189: 968-979

Corresponding Author Name: Laurent Renia and Lisa Ng Fong Poh
Journal Submitted to: EMBO Molecular Medicine
Manuscript Number: EMM-2017-07885